# Clinically translatable quantitative molecular photoacoustic imaging with liposome-encapsulated ICG J-aggregates

Cayla A. Wood [1,2], Sangheon Han [1,3], Chang Soo Kim[1], Yunfei Wen [4], Diego R. T. Sampaio[1,5], Justin T. Harris[6], Kimberly A. Homan[6], Jody L. Swain[7], Stanislav Y. Emelianov [8,9], Anil K. Sood [2,4], Jason R. Cook[6], Konstantin V. Sokolov [1,2,3,10,11✉] & Richard R. Bouchard [1,2,11✉]

Photoacoustic (PA) imaging is a functional and molecular imaging technique capable of high sensitivity and spatiotemporal resolution at depth. Widespread use of PA imaging, however, is limited by currently available contrast agents, which either lack PA-signal-generation ability for deep imaging or their absorbance spectra overlap with hemoglobin, reducing sensitivity. Here we report on a PA contrast agent based on targeted liposomes loaded with J-aggregated indocyanine green (ICG) dye (i.e., PAtrace) that we synthesized, bioconjugated, and characterized to addresses these limitations. We then validated PAtrace in phantom, in vitro, and in vivo PA imaging environments for both spectral unmixing accuracy and targeting efficacy in a folate receptor alpha-positive ovarian cancer model. These study results show that PAtrace concurrently provides significantly improved contrast-agent quantification/sensitivity and $SO_2$ estimation accuracy compared to monomeric ICG. PAtrace's performance attributes and composition of FDA-approved components make it a promising agent for future clinical molecular PA imaging.

[1] Department of Imaging Physics, The University of Texas MD Anderson Cancer Center, Houston, TX, USA. [2] The University of Texas MD Anderson Cancer Center UTHealth Graduate School of Biomedical Sciences, Houston, TX, USA. [3] Department of Bioengineering, Rice University, Houston, TX, USA. [4] Department of Gynecologic Oncology and Reproductive Medicine, The University of Texas MD Anderson Cancer Center, Houston, TX, USA. [5] Department of Physics, University of São Paulo, Ribeirão Preto, Brazil. [6] NanoHybrids, Inc., Austin, TX, USA. [7] Department of Veterinary Medicine and Surgery, The University of Texas MD Anderson Cancer Center, Houston, TX, USA. [8] School of Electrical and Computer Engineering, Georgia Institute of Technology, Atlanta, GA, USA. [9] Wallace H. Coulter Department of Biomedical Engineering, Georgia Institute of Technology and Emory University School of Medicine, Atlanta, GA, USA. [10] Department of Biomedical Engineering, The University of Texas at Austin, Austin, TX, USA. [11]These authors contributed equally: Konstantin V. Sokolov, Richard R. Bouchard. ✉email: ksokolov@mdanderson.org; rrbouchard@mdanderson.org

Molecular imaging is becoming a key tool in both diagnostics and therapy with the emergence of patient-specific treatment schemes to allow for real-time visualization of chemical and biological processes, bridging the gap between traditional in vivo anatomical imaging and ex vivo histopathology[1]. Currently, the only widespread clinical molecular imaging technique is positron emission tomography, which is incredibly sensitive with only trace amounts of radionuclide injections required to achieve high imaging contrast, but it has relatively poor spatial resolution and is highly regulated due to its use of ionizing radiation[2,3]. Of the clinically viable molecular imaging strategies being explored preclinically, photoacoustic (PA) imaging is one of the most promising, making inroads into the clinic due to its nonionizing nature and ability to resolve optical contrast at clinically relevant depths with high spatiotemporal resolution[4,5]. PA imaging has shown encouraging clinical potential using endogenous hemoglobin (i.e., oxy- and deoxyhemoglobin [$HbO_2$ and HHb]) to assess blood oxygen saturation ($SO_2$), an emerging biomarker for many diseases[6–8]. In addition, a variety of exogenous contrast agents (e.g., gold nanoparticles [AuNPs][9] and optically absorbing dyes[10,11]) have shown preclinical potential for molecular imaging of cell-specific markers, allowing for the identification of certain cell types in vivo. Because PA imaging systems often use a conventional pulse-echo ultrasound (US) transducer, PA images can be inherently co-registered with B-mode US images, providing an anatomical reference with functional information[12]. Through concurrent PA–US imaging, it is possible to achieve high-resolution anatomical and molecular imaging at depth with both endogenous and exogenous contrast.

Still, clinical PA imaging is facing formidable challenges. For clinical success, PA imaging systems must deliver accurate quantitative imaging with sufficient spatial resolution to appropriately assess heterogeneity in a given region of interest (ROI). With multi-wavelength PA imaging, endogenous and exogenous absorbers can be identified voxel-wise through the imaging volume via linear spectral unmixing[13] or more complex inversion techniques[14]. In a heterogeneous in vivo environment, however, local fluence variation introduces complex spectral coloring and affects PA-based estimates of chromophore distributions, making robust PA imaging at depths greater than 2 cm a challenge[15]. Such (unknown) fluence variations already make accurate spectral unmixing difficult[16], so the introduction of an exogenous agent with identifying spectral features that overlap with those of hemoglobin further confounds these methods. To mitigate this effect, it is critical that absorption spectra of exogenous agents maintain minimal overlap with the most relevant and dynamic regions of endogenous contrast. Ideally, clinically translatable PA contrast agents should provide high contrast at depth after spectral unmixing from background hemoglobin, maintain molecular specificity to cellular targets, afford sufficient stability during imaging, and have a composition amenable to translation (i.e., biocompatible and scalable for clinical production)[17–19].

A number of inorganic exogenous contrast agents have been investigated for PA imaging. AuNPs gained early popularity as PA contrast agents due to their high absorbance coefficient[20], readily tunable optical properties by varying their size and shape, and silica coating modification that allows for increased in vivo stability and PA signal generation[21,22]. Silver nanoparticles and nanoplates have also been explored, but in vivo stability and toxicity concerns limit their preclinical utility and potential for clinical translation[23]. Ultimately, while many of these agents have demonstrated impressive phantom and preclinical imaging results, they tend to be composed of materials that lack FDA approval, making the barrier to clinical translation relatively high.

Alternatively, organic dyes are commonly used as PA contrast agents as they are widely commercially available[18]. However, many such dyes and dye-conjugates are not particularly well suited for PA imaging, with a molar extinction coefficient that can be four to six of magnitude less than gold-based contrast agents[19], making imaging at depth a serious challenge. Further, most common dyes have absorbance peaks in the 600–800 nm region, which interferes with $SO_2$ quantitation[24]. Theranostic porphysomes have been shown to produce a strong PA signal[25,26] by tight packing of porphyrins inside liposomal membranes; yet, porphyrins are well-known photosensitizers, and thus these agents can interfere with the monitoring of therapeutic interventions[27]. Recently, aggregates formed by nonbiologically active chromophores, such as naphthalocyanines[28], aza-BODIPY[29], and dicarboxyphenyl cyanine[30], were introduced for sensitive PA imaging. In addition, bacterial phytochromes have been used to provide improved PA contrast due to their "photoswitchable" optical absorption[31]. However, these reports did not demonstrate the ability for quantitative, targeted molecular PA imaging in vivo, and the chromophores used in these studies are not FDA-approved, which could significantly delay clinical translation.

One particular dye that has gained significant attention for PA imaging due to its FDA approval status (i.e., for determining hepatic blood flow and for ophthalmic angiography[32]) is indocyanine green (ICG). However, while ICG generates a reasonably strong PA signal, its absorption peak is near the ideal wavelengths for hemoglobin unmixing, making it difficult to spectrally differentiate in vivo and drastically reducing its PA imaging sensitivity. Moreover, the short half-life of ICG in circulation[33] limits its use in disease monitoring or for systemic delivery. Other groups that have explored ICG encapsulation and/or lipid–ICG complexes have used the monomeric form of ICG[34]. The downside of this form is low molar extinction coefficients (in the range of $2.7 \times 10^5 \, \mathrm{cm}^{-1}/\mathrm{M}$ for ICG), which can be hard to detect at depth in tissue with PA imaging.

To address the aforementioned limitations, we introduce a PA contrast agent, PAtrace (Fig. 1a), which is based on liposomal encapsulation of ICG at a high enough concentration for the formation of stable J-aggregates[35]. This configuration results in drastically improved PA signal generation due to increased thermal gradients (from tightly clustered absorbers[22]) and a narrow absorption peak at ~890 nm, which provides increased optical absorption and allows for robust detection sensitivity in the presence of hemoglobin (Supplementary Fig. 1). As shown in Fig. 1b, PAtrace presents with a sharp spectral feature in the 870–920 nm range, where the spectra for hemoglobin remain relatively flat. Conversely, PAtrace's relatively flat absorption spectrum from 760 to 830 nm facilitates accurate PA-based $SO_2$ estimation[12], which is critical for better understanding the dynamics associated with tumor microenvironment regularization[36]. Separation of monomeric ICG from hemoglobin, on the other hand, requires multi-wavelength unmixing of both absorbers in the same wavelength range, confounding PA quantification at varying $SO_2$ percentages and ICG concentrations. In addition, a liposomal formulation of PAtrace allows for PEGylation and conjugation with targeting moieties, improving biocompatibility and conferring specificity against targeting sites, respectively. Molecular targeting of PAtrace is enabled with directional conjugation[37], whereby antibodies are conjugated through the Fc portion, leaving antigen-binding sites on the Fab moiety available for targeting. This diminishes potential nonspecific interactions through Fc receptors that are present in some cells. All of these characteristics allow PAtrace to be a robust agent that makes quantitative, molecular PA imaging in both preclinical and clinical usage more feasible.

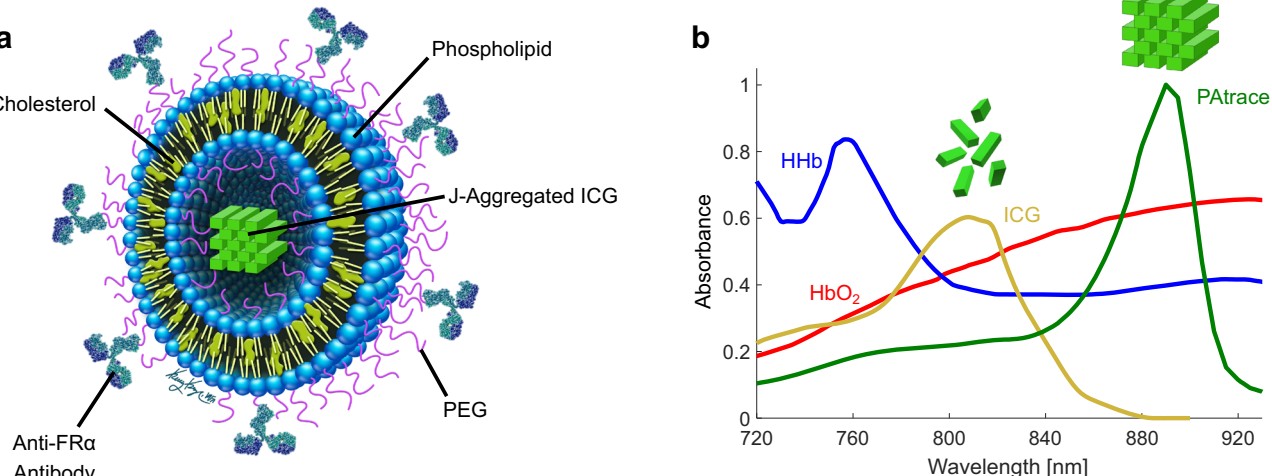

**Fig. 1 PAtrace design. a** Schematic of FRα-targeted PAtrace nanoparticle (FRα-PAtrace). **b** Optical absorbance spectra of endogenous HbO₂ (red) and HHb (blue) with exogenous monomeric ICG in blood serum (gold) and PAtrace (green) demonstrate advantages of unmixing PA images for PAtrace in the presence of blood. Notably, PAtrace's sharp ~890 nm peak facilitates greater spectral separation from hemoglobin, while increased peak optical absorption relative to concentration-matched ICG increases PA signal generation.

In this work, we synthesized, bioconjugated, and characterized liposome-encapsulated ICG J-aggregates (i.e., PAtrace; Fig. 1a) for molecular PA imaging. We then validated PAtrace in phantom, in vitro, and in vivo PA imaging environments for both spectral unmixing accuracy and targeting efficacy with a folate receptor alpha-positive (FRα⁺) ovarian cancer model. For these purposes, we also developed and implemented a surface-fluence-correction model to improve quantification accuracy for in vivo PA imaging.

## Results

**Characterization of PAtrace.** The size distribution, spectrum, loading efficiency, stability in various media, fluence stability, and PA-signal-concentration linearity were evaluated to characterize PAtrace. The size distribution, which was assessed via transmission electron microscopy (TEM), showed a mean (±standard deviation [SD]) particle size of 130 (±37) nm (Fig. 2a, b; Supplementary Fig. 2). Spectral characterization was performed on both intact PAtrace (i.e., J-aggregated form) and ICG released from PAtrace (i.e., monomeric form) with UV–Vis–near-infrared (NIR) spectrophotometry; for this comparison, PAtrace was mixed with Tween 20 at 5% concentration to disrupt the liposomal coating and to dissociate ICG J-aggregates into monomeric ICG molecules. Compared to monomeric ICG, the PAtrace absorbance peak is increased by ~3-fold, narrowed to 30% of the full width at half maximum of ICG, and red-shifted by ~100 nm from the 800 nm monomer peak to the ~890 nm peak of ICG J-aggregates, as shown in Fig. 2c. Note that the absorbance peak varies from 880 to 890 nm between production batches of PAtrace. We also quantified the loading efficiency of PAtrace as the number of ICG dye molecules per nanoparticle. Our data show that PAtrace solution at 1 optical density (OD) has ~8.88 × 10⁸ PAtrace nanoparticles/mL and ~1.37 × 10¹⁵ ICG molecules/mL. Therefore, each PAtrace nanoparticle contains ~1.54 × 10⁶ ICG molecules. This loading efficiency of ICG dyes per liposomal nanoparticle is ~4 × 10² greater than what is reported for liposomes loaded with monomeric IGC dyes[34]. This significant increase in the loading efficiency of PAtrace is most likely associated with dense packing of ICG J-aggregates.

Temporal stability was tested via assessment of the UV–Vis–NIR absorbance spectrum for varying temperatures and solutions at multiple time-points (Supplementary Fig. 3). At the storage condition of 4 °C in phosphate-buffered saline (PBS), PAtrace did not show significant spectral changes through one month, indicating a long shelf life. At 37 °C in 10% fetal bovine serum (FBS), no significant spectral changes were observed over 48 h; however, in 100% FBS at 37 °C, the 890 nm-to-790 nm ratio reduced by 22% after 6 h, 41% after 24 h, and 50% after 48 h, indicating that PAtrace particles degrade progressively and de-aggregate into monomeric ICG in the presence of serum and at physiological temperature. Our in vivo studies (see "Molecular PA imaging of an ovarian cancer model" section) indicate that this stability is sufficient for molecular PA imaging in ovarian cancer models, which require an approximately 30–60 min imaging window. Further, biodegradation of PAtrace could be leveraged for longitudinal measurements with repeated injections of the agent.

Fluence stability was assessed by exposing contrast agents to varying laser-pulse energies during PA imaging. We compared fluence stability (Supplementary Fig. 4) of PAtrace, monomeric ICG, and silica-coated (SiO₂) AuNRs (which are considered the best commercially available contrast agents for PA imaging[21,38]). Each sample was excited at its corresponding maximum absorption wavelength. The average PA signal from 900 laser pulses at each fluence was then normalized by the absorber mass to provide a metric for fluence stability. Nonlinearity in the PA signal with increasing fluence indicates that irreversible changes to the agent were occurring. PAtrace exhibited high fluence stability, presenting a linear PA signal up to ~15 mJ/cm². Conversely, SiO₂ AuNRs started to become nonlinear at >2 mJ/cm². PAtrace also exhibited a fivefold increase in PA signal over free ICG when both had the same dye molarity. Finally, to test the concentration linearity of PA signal generation, multi-wavelength PA imaging was conducted on PAtrace (or ICG) embedded in a gelatin phantom (Supplementary Fig. 5) at concentrations ranging from 25 to 1.6 OD. PA spectra of PAtrace remained constant at varied OD (Fig. 2d, e), while PA signal at the spectral peak (i.e., 880 nm) remained linear with increasing OD (Fig. 2d inset). In contrast, PA spectra of ICG were only constant up to 12.5 OD (Supplementary Fig. 6a, b), and PA signal at the spectral peak (i.e., 810 nm) was linear up to 12.5 OD (Supplementary Fig. 6c) as a significant spectral shift was observed at 25 OD due to aggregation of ICG

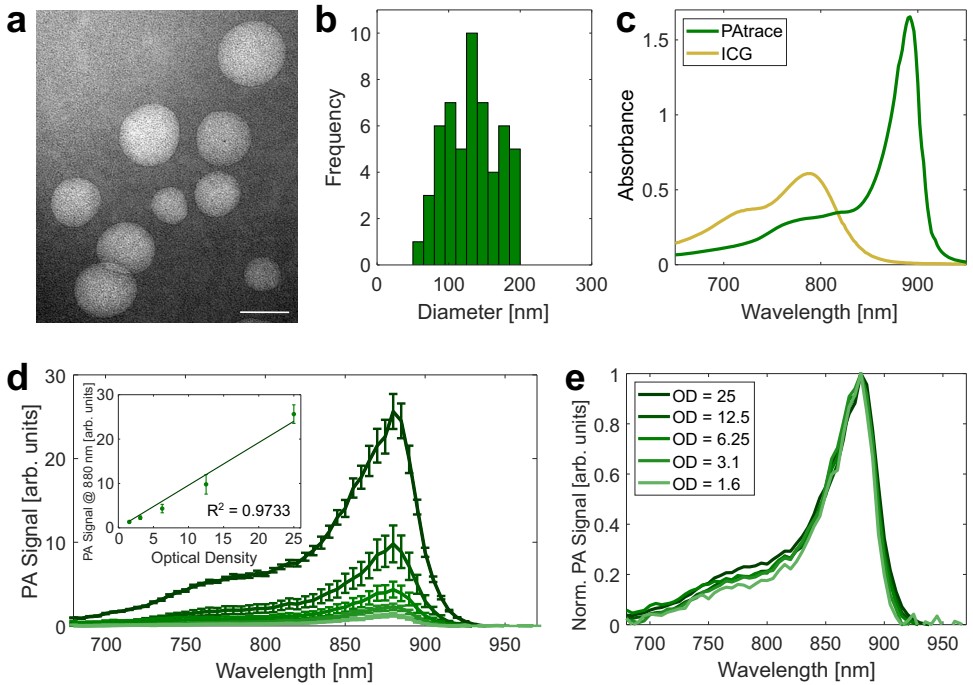

**Fig. 2 PAtrace characterization. a** Representative image of PAtrace TEM (scale bar 100 nm), which measured **b** a size distribution of 130 (±37) nm across all TEM images (N = 5). **c** Spectral characterization of intact PAtrace (green) and ICG released from PAtrace by addition of 5% Tween 20 in water (gold) demonstrates that at equimolar concentrations, PAtrace yields an absorbance spectrum that is red-shifted, narrower, and of increased peak absorbance compared to monomeric ICG. **d** PA imaging of serial dilutions (25–1.6 OD; fluence 0.3–0.9 mJ/cm$^2$) of PAtrace exhibits a constant spectrum over varying concentrations, and (inset) peak PA signal remains linear with increasing OD (error bars indicate mean ± SD across three independent PA acquisitions), while **e** normalization of these PA spectra shows spectral stability (error bars removed for visibility).

molecules (Supplementary Fig. 6a). Note that the absorption peak of monomeric ICG shifts from 790 to 810 nm when it binds to albumin, which is present in blood serum[39].

**Comparison of PAtrace with ICG in phantom and in vivo environments.** To demonstrate the advantages of PAtrace over monomeric ICG for PA signal generation and spectral unmixing in the presence of blood, a polyethylene (PE) tube was embedded in a cylindrical tissue-mimicking phantom (Fig. 3a), and mixtures of PAtrace (or monomeric ICG) and porcine blood were injected into the tube and imaged over multiple wavelengths in the pre-clinical MSOT inVision 256-TF PA imaging system (iThera Medical, Munich, DE), which has a 256-element, 270°-arc acoustic-receiver array with 5 laser-fiber-bundle pairs for PA signal generation. Images were linearly unmixed for hemoglobin (i.e., to assess SO$_2$) and PAtrace (or ICG). Example 800 nm PA images of the phantom are shown in Fig. 3b, with insets showing SO$_2$ and PAtrace (or ICG) images of the tube ROI for 0.9 OD PAtrace (or ICG) with co-mixed 70% SO$_2$ blood. Note that for all PA imaging experiments, "SO$_2$ image" and "PAtrace (or ICG) image" are defined as image matrices that provide measurements of SO$_2$ percentage and relative PAtrace (or ICG) concentration, respectively, based on multi-wavelength unmixing; "SO$_2$ estimate" and "PAtrace (or ICG) signal" are scalars obtained by taking the spatial average of their respective matrices over a given ROI (see "PA post-processing and ROI selection" section). The wavelength combination used for unmixing was determined by a cost function (Supplementary Fig. 7), which included: SO$_2$ estimation error (relative to blood-only samples); error in PAtrace (or ICG) signal ratio (relative to known threefold increase in probe concentration); and number of wavelengths. The cost function optimizes for both the accuracy of PAtrace unmixing

and SO$_2$ estimation, while it includes a modest penalty for the number of unmixing wavelengths used to promote faster acquisition times. The spectral data from each of these multi-wavelength PA acquisitions is shown by the solid lines in Fig. 3e, with the theoretical spectra for these combinations indicated by dashed lines. Theoretical spectra were calculated as the superposition of optical absorption spectra of 70% SO$_2$ hemoglobin and 0.9 OD PAtrace (or ICG); the correlation coefficients between experimental and theoretical spectra were 0.96 and 0.44 for PAtrace and ICG, respectively.

As is shown by both SO$_2$ image insets in Fig. 3b and the plot in Fig. 3c, PA-based SO$_2$ estimation error was significantly worse in the presence of ICG, with two ICG combinations (i.e., 0.9 OD ICG with 50% or 70% SO$_2$) yielding >30% absolute SO$_2$ estimation error; however, it was largely unaffected by the presence of PAtrace, with a maximum absolute error of 18%. The PAtrace signal increased by a factor of 3.3 with a 3-fold increase in concentration, whereas the ICG signal only increased by a factor of 1.4 with a 3-fold increase in concentration (Fig. 3d). The less-than-expected ICG signal increase is not due to the nonlinearity of the monomeric ICG PA signal, which Supplementary Fig. 6c shows to be quite linear at the concentrations used, but rather due to the confounding hemoglobin spectra that are also included while unmixing. Because the absorption peak for monomeric ICG is within the wavelength range (i.e., ~750–850 nm) most sensitive for unmixing HbO$_2$ and HHb, it is difficult to spectrally differentiate ICG from hemoglobin. Thus, some of the increase in PA signal due to an increase in monomeric ICG concentration gets inaccurately assigned to hemoglobin during spectral unmixing.

Visualization of PAtrace and monomeric ICG was compared in vivo with both the inVision and the clinical MSOT Acuity PA imaging systems (iThera Medical). Briefly, matched ICG

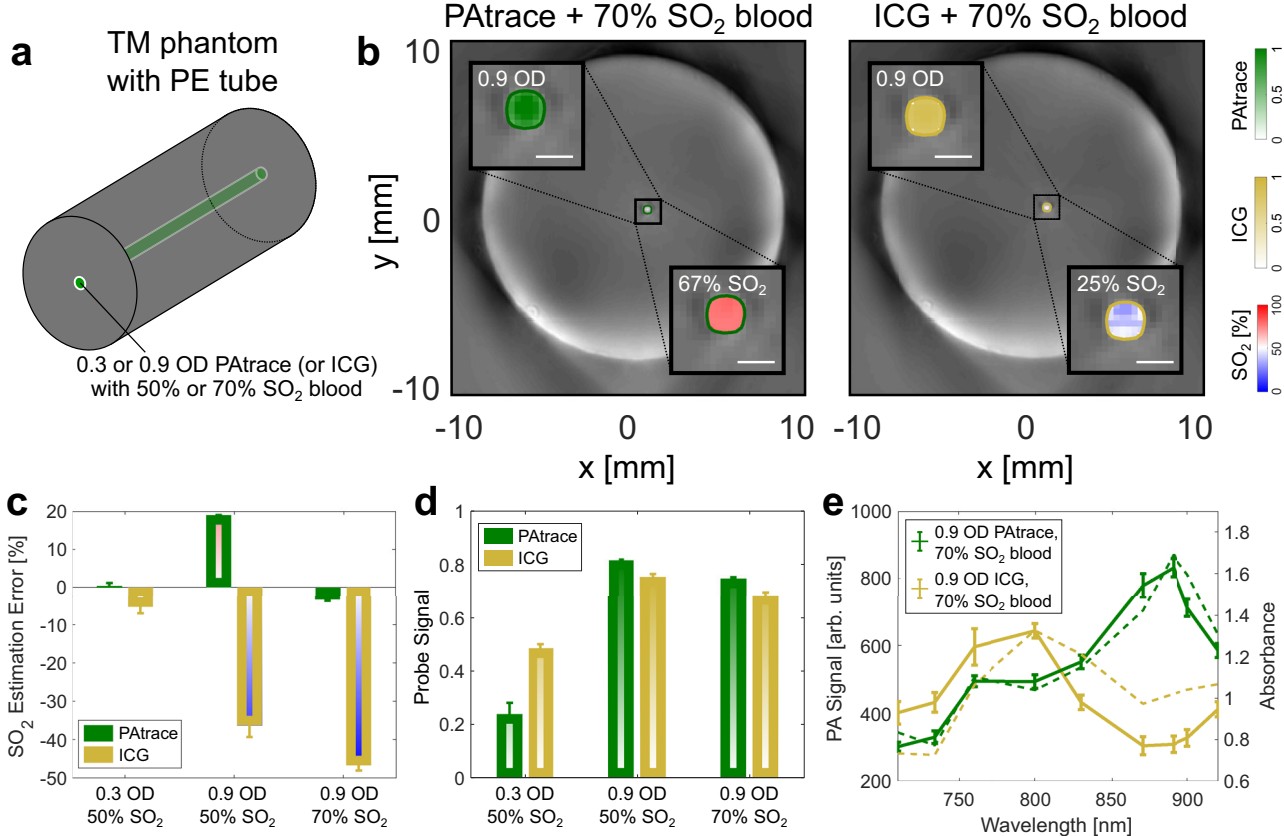

**Fig. 3 Phantom comparison of PAtrace and ICG in blood-containing phantom. a** Schematic of tissue-mimicking (TM) phantom for multi-wavelength PA imaging with an embedded polyethylene (PE) tube for assessment of combinations of PAtrace (or ICG) with blood. **b** 800 nm PA axial images of TM phantom with insets showing $SO_2$ and PAtrace (or ICG) images of the target region for 0.9 OD PAtrace/70% $SO_2$ blood (left) and 0.9 OD ICG/70% $SO_2$ blood (right); color bars for each inset shown on the right. Note that PAtrace and ICG are readily unmixed in their respective insets; however, there is a large error in $SO_2$ estimation in the phantom with ICG, which is not observed in the phantom with PAtrace. White scale bar indicates 0.5 mm. **c** PA-based $SO_2$ estimation error for all concentration/$SO_2$ combinations tested yield <20% absolute error for PAtrace (green bars), while yielding nearly 50% error in presence of ICG (gold bars). **d** PAtrace signal increases by a factor of 3.3 with a 3-fold increase in concentration, whereas ICG signal only increases by a factor of 1.4 with a 3-fold increase in concentration. **e** Spectral data from multi-wavelength PA acquisitions in (**b**) are shown by solid lines (error bars indicate mean ± SD across 12 z-slices), while theoretical spectra for these combinations are indicated by dashed lines. All phantom combinations were fabricated and imaged once.

concentrations of each probe were injected intravenously into a mouse and imaged with the preclinical inVision system. Both unmixed chromophores (i.e., PAtrace or ICG) could be readily visualized in a superficial artery; however, in the aorta (i.e., ~8 mm depth), only PAtrace was visible with matched dynamic range settings (Supplementary Fig. 8). In a separate experiment, matched concentrations of each probe were injected subcutaneously into the mammary fat pad of a mouse and imaged with the clinical Acuity PA–US imaging system. PAtrace was clearly visible at >1.5 cm depth through a tissue-mimicking stand-off, while almost no evidence of monomeric ICG was present for matched molar concentration and injection conditions (Supplementary Fig. 9).

**In vitro evaluation of FRα-PAtrace.** Monoclonal anti-FRα antibodies were modified to contain a thiol group on their gly-cosylated Fc portion using a hydrazide–PEG–thiol linker[37], which allows binding to maleimide groups on the surface of PAtrace via a maleimide-thiol reaction. This directional conjugation[9,37] approach does not involve antigen-binding Fab portions of anti-FRα antibodies, leaving antigen-binding sites fully available for interaction with FRα receptors. FRα-targeted PAtrace (FRα-PAtrace) had the same UV–Vis–NIR spectrum as the maleimide-

functionalized PAtrace (Mal-PAtrace) without antibodies (Supplementary Fig. 10), which indicates that antibody conjugation does not affect the absorbance of PAtrace. Furthermore, FRα-PAtrace exhibited high photothermal stability as there were no changes in its PA spectrum after four consecutive spectral acquisitions (1440 total laser pulses) at a maximum fluence of ~20 mJ/cm² (Supplementary Fig. 11). Conversely, monomeric ICG showed signs of photobleaching as its PA signal intensity decreased after four consecutive spectral acquisitions (Supplementary Fig. 11).

After conjugation, we labeled high-expressing FRα⁺ SKOV3 and low-expressing FRα⁻ A2780 cells with FRα-PAtrace and carried out PA imaging in a 36-well gelatin phantom (Fig. 4c); control cells (i.e., without PAtrace) were included to assess background signal. The 880 nm PA signal from SKOV3 cells incubated with FRα-PAtrace was ~3-fold greater than that from labeled A2780 cells (Fig. 4b, d), which is in good agreement with literature reports of approximately four times higher expression levels of FRα in SKOV3 vs. A2780 cells[40,41]. There was a negligible PA signal from control cells. To assess the cytotoxicity of PAtrace, we tested four cancer cell lines (i.e., ovarian SKOV3 and A2780 cells; breast MDA-MB-468 cells; and head and neck FaDu cells), 3T3 fibroblasts, and normal HUVEC endothelial cells. Cells were incubated with PEGylated PAtrace (i.e.,

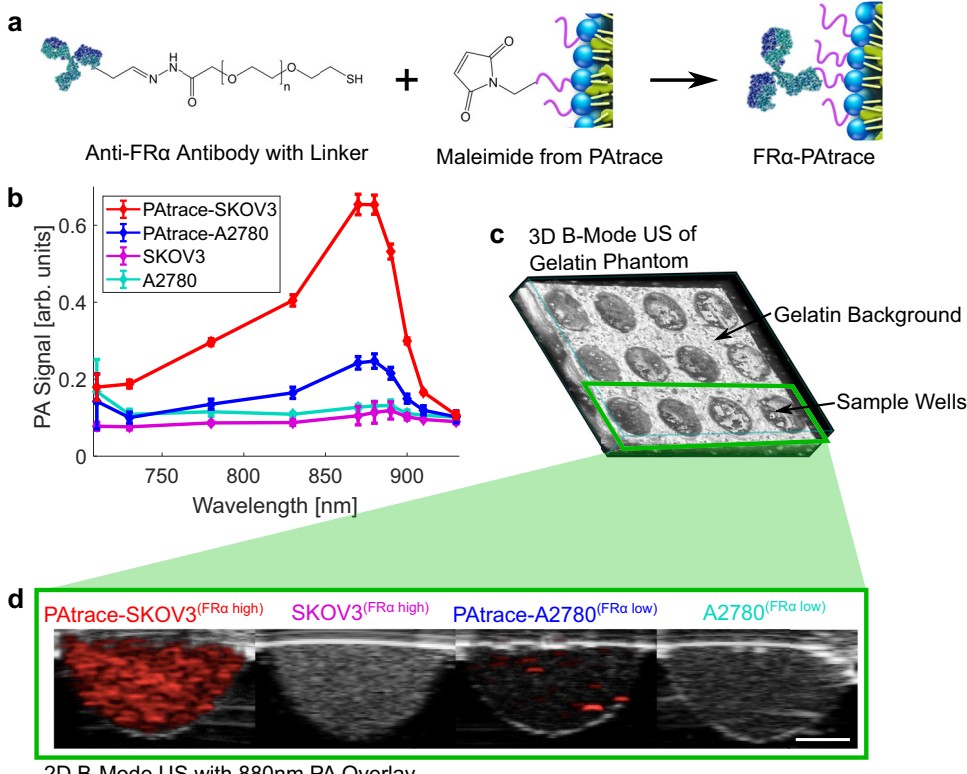

**Fig. 4 In vitro evaluation of FRα–PAtrace targeting. a** Schematic of particle conjugation shows anti-FRα antibody with heterobifunctional linker on Fc moiety attaching to maleimide on PAtrace via directional bioconjugation. **b** PA signal spectra for FRα⁺ SKOV3 (red) and FRα⁻ A2780 (blue) cells labeled with FRα-PAtrace particles with unlabeled SKOV3 (magenta) and A2780 (cyan) as controls (error bars indicate mean ± SD across three independent PA acquisitions/samples). 880 nm PA signal from labeled SKOV3 cells is ~3-fold greater than that from A2780 cells with FRα-PAtrace; there was negligible PA signal from bare cells. **c** Example 3D B-mode US image of 36-well gelatin phantom. **d** Example of PA–US images for cell inclusions show significantly higher 880 nm PA signal in SKOV3 cells than in A2780 cells when both are incubated with FRα-PAtrace. This assay was conducted once with all cell types seeded into three independent inclusions for PA assessment. White scale bar indicates 1 mm.

non-targeted) to avoid any potential biological effects that could be associated with a conjugated antibody. HUVEC cells were chosen as a model of normal cells and were also incubated with ICG molecules and with empty liposomes to assess the cytotoxicity of these individual components. Empty liposomes were evaluated at nanoparticle concentrations ranging from 0.34x to 2.7x of the highest PAtrace concentration (i.e., $\sim 1.57 \times 10^{10}$ nanoparticles/mL, which corresponds to 6.6 OD), and ICG molecules were evaluated using solutions with the same OD values as PAtrace samples. Note that we started with PAtrace (or ICG) at 6.6 OD, which is a particularly high dose that was chosen to intentionally derive toxicity. With PAtrace, cells did not show cytotoxicity below 6.6 OD after 24 h, and only two cell lines showed some cytotoxicity effect (viability > 63%) below 6.6 OD after 48 h compared to controls (Supplementary Fig. 12). Empty liposomes did not exhibit any cytotoxicity in HUVEC cells, while ICG showed some cytotoxicity in HUVEC cells at high ICG concentrations of 3.3 and 6.6 OD for both time-points (Supplementary Fig. 13). These data indicate that the observed minor cytotoxic effect at high PAtrace concentrations might be associated with a very high loading of ICG dye molecules inside PAtrace.

**Molecular PA imaging of an ovarian cancer model**. Mice with orthotopic SKOV3 ovarian tumors were imaged in the inVision system when tumors reached ~5 mm in diameter, as confirmed by magnetic resonance imaging (MRI) (Fig. 5a). Preinjection images (Fig. 5b, top-left) were acquired with the established

PA-imaging presets (Fig. 5b, bottom-left) for each wavelength at z-positions encompassing the liver, spleen, kidney, and tumor. Targeted FRα-PAtrace or non-targeted RG-16-PAtrace was injected intravenously (Fig. 5b, top-middle), and 3D PA imaging was repeated at multiple time-points postinjection (Fig. 5b, top-right). Monoclonal RG-16 antibodies were used as a non-targeted control because they do not interact with any mouse or human epitopes. PA images were reconstructed in the ViewMSOT software (iThera Medical), then imported to MATLAB R2019b (MathWorks, Natick, MA) for post-processing. This included a surface-fluence correction to account for variation in the inVision irradiation field, which was modeled from the ring geometry to include wavelength-dependent water absorption and approximate beam divergence (Supplementary Fig. 14). To smoothly propagate the surface-fluence correction through the volume, the mouse surface was manually segmented, and a finite element method mesh was generated using average tissue parameters, which were set to be constant across wavelength. Fluence-corrected PA images were then linearly unmixed for PAtrace (Fig. 5c) and hemoglobin (i.e., to assess SO₂; Fig. 5d) with a wavelength combination chosen by a cost function (Supplementary Fig. 15), which included change between SO₂ estimates from preinjection to postinjection, preinjection PAtrace signal, and number of wavelengths; then, 3D ROIs were manually ascribed to the tumor and liver volumes based on MRI (Fig. 6a, d) and preinjection 800 nm PA imaging (Fig. 6b, e) data. Postinjection PAtrace accumulation was quantified by PAtrace contrast enhancement (CE) relative to preinjection (see "PA post-processing and ROI selection" section). SO₂ and PAtrace CE images were compared

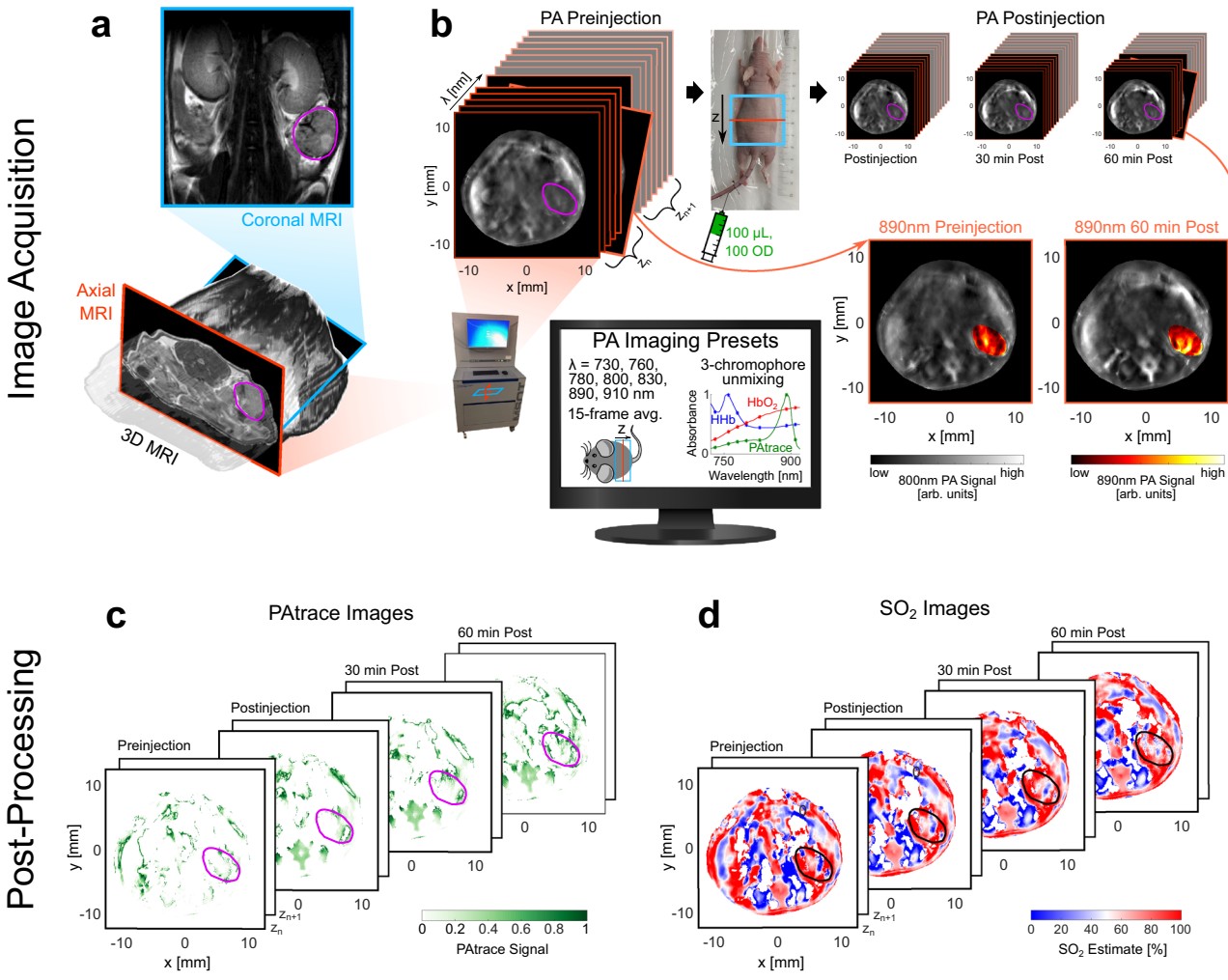

**Fig. 5 Summary of methods for in vivo PA imaging study. a** Coronal (top) and axial (bottom) MR images to monitor tumor growth, and 3D (bottom) MRI data to localize ovarian tumors in PA images. **b** PA imaging workflow: mice placed in the inVision system (bottom-left), and system set to PA imaging presets (i.e., wavelengths, averaging; bottom-left). Preinjection (top-left) PA axial images (800 nm shown) were acquired for each wavelength at z-positions encompassing the liver, spleen, kidney, and tumor. 100 μL of either targeted FRα-PAtrace or non-targeted RG-16-PAtrace (100 OD) was injected intravenously; photograph (top-middle) of mouse setup for intravenous PAtrace injection with a blue box showing PA scan length and orange line indicating current axial slice. Postinjection (top-right) imaging was performed at three time-points, allowing for slice-by-slice co-registration. Representative 800 nm (grayscale colormap) anatomical image with overlaid 890 nm (hot colormap) PA images of the matched tumoral cross-section are shown preinjection (bottom-middle) and 60 min postinjection (bottom-right) to visualize signal enhancement after PAtrace has accumulated. Images were fluence corrected and linearly unmixed for **c** PAtrace and **d** hemoglobin (to obtain SO₂) for each z-position and time-point.

voxel-wise across all time-points to capture local changes through time.

As seen in Fig. 6c, f, substantial PAtrace CE from both targeted and non-targeted PAtrace appeared at the MRI-confirmed tumor location immediately post injection. This PAtrace CE is most likely associated with the agent in the blood pool. Thirty min postinjection, however, PAtrace CE was significantly greater (Fig. 6h, right; $p = 0.017$) for FRα-PAtrace than for RG-16-PAtrace, and this persisted at 60 min postinjection ($p = 0.044$), indicating molecular specificity of the agent. FRα expression in SKOV3 tumors was confirmed by Western blotting of excised tumor tissue (Supplementary Fig. 16). PAtrace CE associated with both FRα-PAtrace and RG-16-PAtrace accumulation was observed in the liver (Fig. 6h, left) and spleen (Fig. 6h, middle) across all time-points and did not significantly differ between targeted and non-targeted PAtrace; this is an expected result for nonspecific uptake of nanoparticles with similar composition by the reticuloendothelial system. There was no PAtrace CE observed in the kidneys at any time-points, indicating that

PAtrace is not cleared renally, as has been observed with particles of monomeric ICG encapsulated in liposomes[42]. SO₂ was also assessed across all time-points to ensure that no significant physiological changes occurred during imaging. As shown in Fig. 6g, SO₂ estimates in the liver (left), spleen (middle), and tumor (right) did not change significantly through time for a given mouse. Although there was significant inter-tumor SO₂ variation observed, which is expected due to variations between tumors[43], there existed relatively low longitudinal SO₂ variation for matched tumor comparisons.

**In vivo biosafety and circulation half-life of PAtrace**. Wild-type mice were injected with FRα-PAtrace and imaged in the inVision system to determine the circulation half-life. Then, 24 h post-injection, mice were sacrificed and processed for hematology, blood chemistry, and histology. To measure circulation half-life, mice were imaged continuously through the neck before and for 15 min after intravenous injection of FRα-PAtrace. The 890 nm

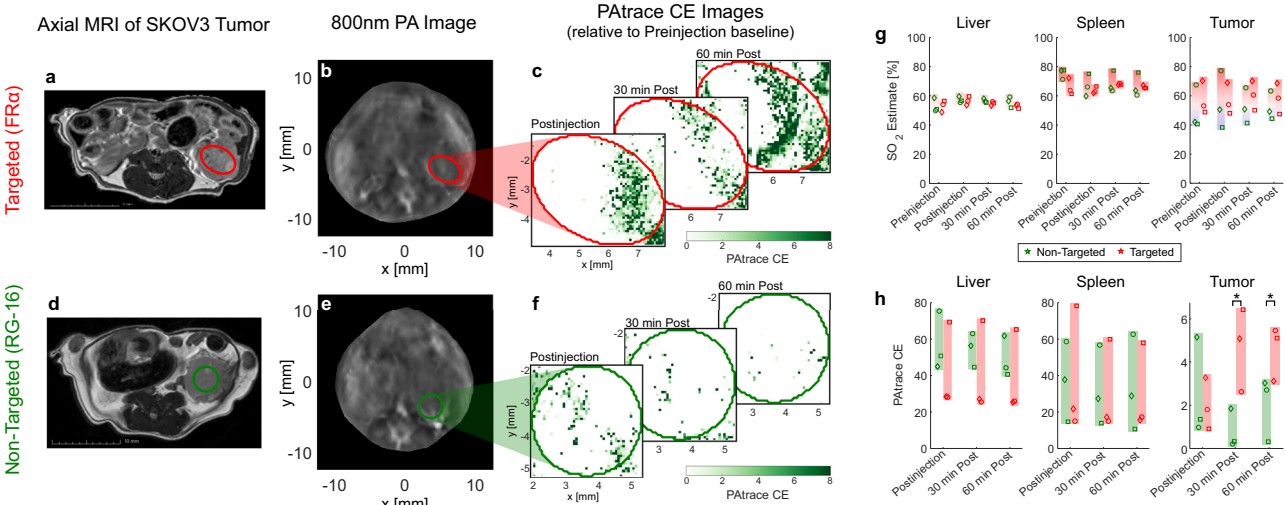

**Fig. 6 In vivo PA imaging specificity of targeted FRα-PAtrace in orthotopic ovarian cancer model.** Results for targeted FRα-PAtrace (red) and non-targeted RG-16-PAtrace (green). Representative **a, d** axial MRI and **b, e** 800 nm PA image of SKOV3 tumors with ROIs indicated by red/green ellipses. MRI volumes were used to establish the position and size of tumors in PA images for ROI placement. Representative PAtrace contrast enhancement (CE) images thru postinjection time-points of **c** targeted FRα-PAtrace and **f** non-targeted RG-16-PAtrace. **g** PA-based $SO_2$ estimates for liver (left), spleen (middle), tumor (right) ROIs show that $SO_2$ does not change significantly through time, indicating no substantial physiological changes occurred during imaging. **h** PAtrace CE in the liver (left) and spleen (middle) ROIs suggest no significant difference between FRα-PAtrace and RG-16-PAtrace liver or spleen accumulation. PAtrace CE in tumor ROIs (right) presents positive enhancement for both targeted and non-targeted PAtrace immediately postinjection, but with targeted enhancement significantly greater 30 min ($p = 0.017$) and 60 min ($p = 0.044$) postinjection. Error bars indicate mean ± SD across three independent subjects.

signal from a superficial surface vein was plotted across time and fit to an exponential to yield a circulation half-life estimate of 10 (±3.5) min (Supplementary Fig. 17). As was determined by a board-certified veterinarian, hematology (Supplementary Table 1) and blood chemistry (Supplementary Table 2) results show no evidence of toxicity as a result of FRα-PAtrace injection. In addition, no morphologic changes related to the FRα-PAtrace injection are observed in H&E samples of liver, spleen, kidney, and heart tissue (Supplementary Fig. 18).

## Discussion

This project was started with the goal of developing a robust PA contrast agent based on particular specifications that need to be achieved for a successful clinical translation, including (i) high absorbance in the NIR optical tissue window; (ii) the possibility of quantitative combined functional (e.g., $SO_2$) and molecular PA imaging; (iii) stability during storage and in vivo applications; (iv) molecular specificity; and (v) reduction of regulatory approval burden. Our agent, PAtrace, was built around ICG dye because of its optical absorbance in the NIR and its FDA approval status. Then, PAtrace's absorbance and PA signal generation were significantly increased by forming J-aggregates that allowed sensitive detection at clinically relevant depths. Further, shifting the absorbance peak to ~890 nm eliminated overlap with distinctive spectral features of hemoglobin that are commonly used for blood oxygenation measurements. In phantom and in vivo settings, we achieved accurate $SO_2$ estimation and PAtrace signal (e.g., approximately threefold increase with commensurate concentration increase) in the presence of blood due to the inherent separation of hemoglobin and PAtrace spectral features. During systemic delivery, an exogenous agent will often present with a similar (or lower) intravoxel optical absorption to that of hemoglobin[44]. Because of its sharp absorption peak at ~890 nm, PAtrace lies in a spectral region where hemoglobin absorption spectra are relatively flat. This inherent spectral separation allows

for straightforward unmixing of hemoglobin and PAtrace with high in vivo contrast. In turn, these image data require only a simple post-processing strategy, which corrects for surface-fluence variation and removes voxels below baseline variation, to provide accurate and concurrent quantification of both hemoglobin and PAtrace.

ICG J-aggregates in PAtrace were stabilized by liposomal encapsulation with decades of successful applications in liposomal drug delivery formulations[45]. PAtrace has a mean diameter of 130 nm, is stable under laser fluences up to 15 mJ/cm², shows a linear PA response over concentrations up to 25 OD, and is very stable for 6 h in serum and for at least one month under storage conditions. Furthermore, the use of functional maleimide lipids in the formulation of PAtrace enabled robust directional conjugation of monoclonal antibodies for molecularly specific targeting of tumor cells that overexpress FRα. To this end, we demonstrated successful targeting of PAtrace to the FRα receptor on SKOV3 ovarian cancer cells in both in vitro and in vivo settings and observed increased PAtrace CE of targeted FRα-PAtrace in SKOV3 tumors relative to non-targeted RG-16-PAtrace. Because we demonstrated PAtrace stability of at least 6 h in serum and saw significant uptake of FRα-PAtrace 60 min postinjection, probe stability is sufficient for our imaging window. Imaging of FRα in ovarian cancer cells has important implications in developing molecular therapies for FRα blockade and for longitudinal therapy monitoring in the clinic[41,46].

In this study, we successfully image a specific molecular target using a PA contrast agent that is based on liposome-stabilized dye aggregates. This is significant because dye aggregates provide superior PA-signal-generation ability and stability relative to individual dye molecules, two critical features that can be leveraged for improved imaging sensitivity and depth penetration. Further, absorbance peaks of dye molecules that are commonly used in preclinical studies and clinical trials significantly overlap with distinctive spectral features of hemoglobin that, as our study demonstrates, can adversely affect the quantitation of PA imaging

data. However, it must be noted that unmixing accuracy is still negatively affected by spectral coloring, and thus the application of local-fluence correction (i.e., not simply surface-fluence correction, as was applied in this work) may be necessary for the relatively deep ROIs likely to be encountered for in-human applications. In addition, further improvements in the imaging sensitivity of clinical PA transducers, which typically have a reduced receive-angle (e.g., linear array) compared to the MSOT inVision's arc array, may be required to achieve clinical performance commensurate with the preclinical results shown herein.

We carefully selected the composition of PAtrace to simplify future clinical translation. Every major component in the core formulation (i.e., DPPC, DSPE-PEG, DSPE-PEG-Maleimide, cholesterol, ICG, and anti-FRα antibody) can be obtained from sources compliant with current Good Manufacturing Practices (cGMP), which is one major hurdle in clinical translation of a product. Another major barrier to the clinic is the toxicity profiling of each of the components. If a component is not classified as Generally Regarded as Safe or has not had toxicology or pharmacology studies under current Good Laboratory Practices (cGLP), then an extensive amount of preclinical testing is required before clinical testing can occur[47]. Each of PAtrace's components either has been already used in a clinical trial or is currently approved by the FDA. DPPC has made it to Phase III clinical trials in the formulations Stimuvax (Oncothyreon, Inc., R Flourney, CA) and ThermoDox (Celsion Corporation, Lawrenceville, NJ)[48]. DSPE-PEG has been featured in both FDA-approved Doxil® (Sequus Pharmaceuticals, Inc., Menlo Park, CA) and Onivyde™ (Merrimack Pharmaceuticals, Cambridge, MA)[48]. DSPE-PEG-Maleimide is currently featured in Phase I clinical trial of MCC-465 (Mitsubishi Tanabe Pharma Corporation, Osaka, JP)[48]. Cholesterol is included in the majority of clinical liposome agents, including Doxil®, DaunoXome® (NeXstar Pharmaceuticals, Boulder, CO), and others[48]. ICG has been FDA approved since 1959[49]. Finally, anti-FRα antibodies have been studied in Phase III clinical trials under the name farletuzumab (Morphotek, Inc., Exton, PA)[50]. Overall, given the clinical history of these components and their relatively low toxicities, we would expect a streamlined path for PAtrace through cGMP, cGLP, and a Phase 1 clinical trial.

To use this technology in monitoring disease progression and treatment response, it will be essential to perform PA imaging with repeat injections at multiple time-points. Currently, the gold standard for monitoring disease progression is through observation of anatomical changes, such as tumor volume. However, molecular changes—such as an increase in cell receptor expression—have been shown in many studies to be earlier markers of disease progression, allowing for more timely treatment strategies[2]. The non-ionizing nature of PA imaging makes repeated acquisitions feasible, while its limited in vivo stability (i.e., significant spectral changes observed at 48 h in serum) makes PAtrace amenable to repeated injections for longitudinal monitoring of disease or response to therapy. Although this study investigates PA imaging of PAtrace for FRα⁺ SKOV3 ovarian cancer, this technology can be applied to a variety of cell receptors (e.g., epidermal growth factor receptor) and cancer cells that are accessible with PA imaging. For example, FRα-PAtrace could be used to image triple-negative breast cancer, a subtype of invasive breast cancer with a particularly poor prognosis and that has been shown to overexpress FRα[51–53].

In addition, liposomes have been extensively researched as nanocarriers for drug delivery as they allow for targeted delivery of chemotherapeutics while reducing toxicity to normal organs[54], with a few such formulations having already made it into the clinic (e.g., PEGylated liposomal doxorubicin, such as Doxil®, for various cancer treatments)[45]. However, one of the key roadblocks in the development of such next-generation, liposome-mediated drug delivery is determining how each liposomal formulation will affect blood circulation, biodistribution, biodegradation, and toxicity[45]. Because of our agent's liposomal formulation, it is feasible that PAtrace could be used as a nearly identical surrogate to predict the pharmacokinetics of liposome-encapsulated therapeutics with high accuracy[55,56]. Further, it has been recently demonstrated in a subcutaneous tumor model that liposomes with encapsulated monomeric ICG can be used to visualize probe extravasation in real-time using PA imaging[57]; this study suggests that it may also be possible to encapsulate a therapeutic agent in PAtrace and use PA imaging to track its delivery to the desired ROI[58] or to monitor the agent extravasating in real-time.

In conclusion, we have developed and validated a PA contrast agent, PAtrace, starting from a set of aforementioned clinically critical specifications. PAtrace provides strong PA signal generation that can be readily differentiated from endogenous hemoglobin, allowing for high-contrast molecular imaging combined with quantitation of physiological parameters, such as blood oxygenation. PAtrace can be conjugated with targeting moieties for cell-specific imaging and can be composed of all FDA-approved components. Overall, PAtrace has a combination of properties that can facilitate the emergence of molecular PA imaging as a real-time imaging modality for in vivo visualization of molecular and functional processes in a clinical setting.

## Methods

**Composition and synthesis of PAtrace**. PAtrace was formed by a combination of lipid film hydration, sonication, and membrane extrusion methods. A combination of 1,2-dipalmitoyl-sn-glycero-3-phosphocholine (DPPC), 1,2-distearoyl-sn-glycero-3-phosphoethanolamine-N-[methoxy(polyethylene glycol)-2000] (DSPE-mPEG), 1,2-distearoyl-sn-glycero-3-phosphoethanolamine-N-[maleimide(polyethylene glycol)-2000] (DSPE-PEG-Mal), and cholesterol (Avanti Polar Lipids, Alabaster, AL) was added to a 100 mL, round-bottom flask at a molar ratio of 90:7.5:1:1.5, respectively. One hundred mL of chloroform was used to solubilize the 100 mg lipid mixture, then a lipid film was produced inside the flask by rotary evaporation at 40 °C. Once the film was produced, a 100 mL solution of United States Pharmacopeia Reference Standard grade ICG (MilliporeSigma, Burlington, MA) at a concentration of 1.0 mg/mL in a 10 mM 2-(N-morpholino)ethanesulfonic acid (MES) buffer, pH 5.0 was added to the flask. After hydration of the lipids, the solution was sonicated using a probe sonicator equipped with a microtip (Branson Ultrasonics Corp., Danbury, CT) at the lowest power setting and a 10% duty cycle for 60 min. During sonication, the flask was partially submerged in a temperature-controlled water bath set to 40 °C. Then, sodium azide was added at a concentration of 0.01% to inhibit potential bacterial growth. The solution was capped and stored in the dark until the absorbance at 890 nm was >3× the absorbance at 780 nm, which was approximately 60 days.

Once the ICG J-aggregates formed, the solution was passed through a 0.45 μm PVDF filter, followed by a 0.22 μm PVDF filter (MilliporeSigma). The 100mL solution was then dialyzed using 300 kDa dialysis tubing (Spectrum Labs, San Francisco, CA) against 10 L of 10 mM MES buffer, pH 6.0, for 5 days at 4 °C. This step has been repeated a minimum of 3 times, and the liquid which permeated through the centrifugal filter was measured by a Cary 60 UV–Vis–NIR spectrophotometer (Agilent Technologies, Inc., Santa Clara, CA) to confirm the removal of free ICG. The final PAtrace suspension was then concentrated using a 100,000 MWCO centrifuge filter (MilliporeSigma) in a swinging bucket centrifuge (Eppendorf, Enfield, CT) operating at 1500g. The concentrated PAtrace solutions were then passed through a 0.45 μm PVDF filter, followed by a 0.22 μm PVDF filter. Finally, the solutions were extruded 12 times through a 100 nm-pore-size polycarbonate membrane (Avanti Polar Lipids) using an extruder (Avanti Polar Lipids) with two 1 mL glass syringes. The Mal-PAtrace was stored at 500 OD at 4 °C until use.

To determine the number of ICG molecules per PAtrace nanoparticle, the number concentration of PAtrace was measured by tunable resistive pulse sensing[59,60] using a qNano Gold system (Izon Science Ltd., Oxford, UK). A PAtrace suspension was first filtered through a 0.45 μm PVDF syringe filter (CELLTREAT, Pepperell, MA) and diluted ~100 times in the injection buffer (Dulbecco's PBS with wetting solution concentrate; Izon Science Ltd.) to be ~5 OD, then added to the qNano Gold system equipped with an NP150 nanopore membrane. The concentration of nanoparticles was determined based on their flow rate through the membrane under 5 mbar of pressure. To measure the concentration of ICG dye molecules, the PAtrace sample was mixed with ethanol to achieve a 1:1 v/v ethanol/water ratio. Under this condition, the liposomal coating is dissolved, and ICG J-aggregates dissociate to monomeric ICG molecules almost instantly. Then, the absorbance of the ICG solution at 790 nm was measured using

a Synergy HT Microplate Reader (BioTek Instruments, Winooski, VT), and the concentration of ICG molecules was determined using a linear calibration curve of ICG ($R^2 = 0.999$) in 1:1 v/v ethanol/water solution.

**Directional bioconjugation of PAtrace.** Anti-FRα monoclonal antibodies (clone 548908, Thermo Fisher Scientific Cat# MA5-23917) and anti-RG-16 monoclonal antibodies (clone RG-16, Sigma-Aldrich Cat# I0138) were directionally conjugated to PAtrace containing maleimide reactive thiol groups using a heterobifunctional linker[37]. The linker consists of a polyethylene glycol (PEG) chain terminated at one end by a hydrazide moiety and at the other end with a thiol group—hydrazide–PEG–thiol linker (PG2-HZTH-3k; Nanocs, New York, NY). The glyco-sylated Fc region of the antibodies was first mildly oxidized to form aldehyde groups at the terminal carbohydrate moieties. Anti-FRα monoclonal antibodies were received as 0.1 mg of lyophilized powder from the manufacturer and were reconstituted in 100 μL of 100 mM sodium phosphate buffer (pH 7.5) to achieve a concentration of 1 mg/mL. Anti-RG-16 monoclonal antibodies were received as a 0.2 mL solution at a concentration of 13.1 mg/mL (IgG₁ isotype, as determined by the manufacturer using radial immunodiffusion). To reconstitute anti-RG-16 antibodies in a phosphate buffer, 7.6 μL of the stock solution was first added to 3 mL of 100mM sodium phosphate buffer (pH 7.5). The solution was then transferred to a 100,000 MWCO centrifugal filter (MilliporeSigma) and centrifuged at 3100*g* for 20 min. The remaining antibodies were reconstituted in 100 μL of 100 mM sodium phosphate buffer, pH 7.5, to achieve the final concentration of 1 mg/mL of IgG₁ isotype. In a typical reaction, a 10 μL aliquot of 100 mM sodium periodate was added to the 100 μL antibody solution to achieve ~1700 molar excess for 30 min. The oxidation reaction was quenched by the addition of PBS at 10-fold excess v/v. Then, 20 μL of the hydrazide–PEG–thiol linker, stored at 46.5 mM in EtOH solution (EtOH:H₂O 1:1 v/v), was added to 6 mL of the oxidized antibody solution (~170 molar linker excess) for 30 min. The unreacted linker was removed by two purification steps using a 10,000 MWCO centrifugal filter (MilliporeSigma) at 3100*g* for 20 min. The purified thiol-modified antibodies were reconstituted in 40 mM HEPES (pH 8.4) at 1 mg/mL and then were mixed with Mal-PAtrace suspension at 500 OD in 10 mM MES buffer, pH 6.5, at a 10:4 volume ratio. The pH value was adjusted to pH 8, and the conjugation reaction was allowed to proceed for 1 h at room temperature. During the conjugation step, a stable thioether linkage bond is formed between PAtrace and antibodies. Unreacted free antibodies were removed by aspiring supernatants after centrifugation of the mixture at 17,200*g* for 30 min twice. The resulting conjugates were stored in sterilized PBS at 100 OD and 4 °C.

**TEM of PAtrace.** PAtrace (10 μL, 10 OD) was placed on 100-mesh copper grids coated with carbon and formvar and pretreated with poly-L-lysine for ~1 h, then negatively stained with Millipore-filtered aqueous 2% uranyl acetate. The stain was blotted dry from the grids with filter paper, and the samples were allowed to dry. Samples were then examined on a JEM-1010 transmission electron microscope (JEOL USA, Inc., Peabody, MA) at an accelerating voltage of 80 kV. Digital images were obtained using the AMT Advantage HR/HR-B CCD Camera System (Advanced Microscopy Techniques, Corp., Danvers, MA). The size distribution of PAtrace was assessed on TEM images using ImageJ 3.0 software, which provides a digital caliper tool[61]. First, the scale was set according to the TEM image scale bar. Then, the diameter of each PAtrace nanoparticle in the TEM images was measured using digital calipers. Fifty-four PAtrace nanoparticles from five independently acquired TEM images were used to determine the PAtrace size distribution.

**Spectral comparison of PAtrace to monomeric ICG.** PAtrace was centrifuged at 50*g* for 10 min to remove any potential aggregates. After centrifugation, the supernatant was collected and measured with UV-Vis-NIR spectrophotometry; PAtrace was diluted to 2 OD at 890 nm with PBS buffer for the UV–Vis–NIR measurement. Then, ICG was released from PAtrace by mixing 200 μL of 5% Tween 20 (MilliporeSigma) in PBS with 200 μL of the 2-OD stock PAtrace; this spectrum was compared with 200 μL of 2-OD stock PAtrace mixed with 200 μL PBS. After mixing for 30 min, absorbance spectra were acquired in a Synergy HT Microplate Reader (300–950 nm; 2 nm step size) using Gen5 software. All measurements were carried out in triplicate. Note that all UV–Vis–NIR absorbance spectra were measured with a path length of 1 cm.

**Stability of PAtrace in different media.** The stability of the PAtrace absorbance spectrum was tested for varying temperatures (4 and 37 °C) and solutions (PBS, 10% FBS in DMEM, and 100% FBS) at multiple time-points (6 h, 24 h, 48 h, 1 week, and 1 month). Triplicate samples were incubated for each condition at 2 OD. Absorbance spectra were collected in a Synergy HT Microplate Reader (400–950 nm; 2 nm step size) using Gen5 software. Triplicate absorbance values were averaged, and spectra were plotted for each condition.

**Fluence stability measurements.** Samples were injected into a glass tube and sub-merged in water to facilitate coupling with the US transducer. PA images and spectra were acquired with the Vevo 2100 LAZR high-frequency PA–US imaging system (FUJIFILM VisualSonics, Toronto, Canada) using a linear array transducer (LZ250; 21 MHz center frequency) with a 2.4 cm field of view and a nominal 5 Hz frame rate. Images were acquired to determine the spectral peak of each agent (680–970 nm; 37 dB gain; persistence of 8). Each sample was then exposed to 900 laser pulses at its respective maximum-PA-signal wavelength for seven different fluence conditions (1, 2, 5, 10, 15, 20, and 25 mJ/cm²). To determine the stability of the nanoparticles, the PA signal at each fluence from all pulses was averaged and plotted vs. fluence.

**Photothermal stability measurements.** To assess the photothermal stability of FRα-PAtrace and monomeric ICG, 20 μL of either FRα-PAtrace in PBS or ICG in 50% EtOH was injected into a 0.4mm-inner-diameter PE tube (Becton, Dickinson and Company, Franklin Lakes, NJ). Each tube was then imaged with the Vevo 2100 LAZR system (680, 730, 760, 780, 790, 810, 830, 850, 870, 880, 890, 900, 920, 940, and 970 nm; 40 dB gain; persistence of 6). The spectral scan was repeated four times, which exposed the sample to 1440 pulses over the PA spectral acquisition. An ROI was manually placed in the center of the tube, and the PA signal was averaged within the ROI. The imaging and analysis procedure was repeated across four tubes. For each tube, the four spectra were normalized by the mean of the first spectrum's PA values. The mean (±SD) was then calculated across the four tubes' normalized spectra, and the first and last spectra were plotted versus wavelength for comparison, while the agreement between these spectra was assessed with a correlation coefficient.

**Studies in multi-well gelatin phantoms.** To assess the linearity of PA signal generation, PAtrace (or ICG) was placed in a gelatin phantom with multiple sample wells[62,63]. Briefly, 36 semi-ellipsoidal inclusions were designed with Fusion 360 software (Autodesk, Inc., Mill Valley, CA), each with a volume of ~57 μL. Then, polydimethylsiloxane (Dow Corning, Midland, MI) was used to create a 3D-printed mold. The phantom background was comprised of 8% w/v 225-Bloom, Type B gelatin, 1% v/v propanol and 0.1% v/v glutaraldehyde (MilliporeSigma). PAtrace (or ICG in 4% w/v bovine serum albumin in water) was serially diluted from 25 to 1.6 OD, then each concentration was mixed with a 16% w/v gelatin solution in a 1:1 v/v ratio before decanting into inclusions in the phantom. All samples were prepared in triplicate. A final ~3 mm layer of gelatin was deposited atop the inclusions to immobilize the cells in each well.

The same phantom type was used for in vitro PA cell imaging studies. SKOV3 (FRα⁺) and A2780 (FRα⁻) cells were seeded at $1 \times 10^6$ cells/well in a 6-well cell-culture plate and incubated overnight. PAtrace, conjugated with either non-targeted RG-16 or targeted anti-FRα antibodies, was then added at 3.5 OD per well and incubated for 2 h at 37 °C. Unbound particles were removed by washing with PBS; cells were removed by trypsinization and collected by centrifugation at 300*g* for 10 min. The resulting cell pellets were reconstituted in ~100 μL of 4% v/v paraformaldehyde solution (Electron Microscopy Sciences, Hatfield, PA) in PBS. Reconstituted cells were then mixed with 16% w/v gelatin in a 1:1 v/v ratio before transferring to gelatin phantom wells. A final ~3 mm layer of gelatin was deposited over the top of the inclusions. All samples were prepared in triplicate.

For all multi-well gelatin phantoms, volumetric B-mode US and multi-wavelength PA images were acquired with the Vevo 2100 LAZR system (710, 730, 780, 830, 870, 890, 900, 910, and 920 nm; 0.47 mm step size; 40 dB gain; persistence of 6). The average PA signal in each inclusion was obtained through volumetric segmentation of B-mode US images at the same dynamic range. Laser fluence ranged from 0.3 to 0.9 mJ/cm² in the analyzed region at the center of the sample.

**Tissue-mimicking phantom comparison of PAtrace to monomeric ICG.** Experiments were conducted in a tissue-mimicking phantom to demonstrate the advantages of PAtrace over monomeric ICG in accurate quantification of PA imaging data with SNR that would begin to emulate the in vivo condition at depth. To this end, a 0.4 mm-inner-diameter PE tube was embedded in a 2 cm diameter 8% m/m 225-Bloom, Type B gelatin cylinder with 0.6% m/m silica, 0.01% m/m India ink, and 8% m/m Intralipid added, which was based on the previous work[62]. Heparinized porcine blood (Animal Technologies, Tyler, TX) was prepared at specific SO₂ percentages[24] and mixed with PBS at a 9:1 ratio to approximate a 10% v/v of blood. Mixtures of PAtrace (or ICG) with blood were then prepared to achieve 0.3/0.9 probe OD and 50%/70% blood SO₂, injected into the tube, and imaged in the inVision system (710, 734, 760, 800, 830, 870, 890, 900, and 920 nm; 10-frame averaging). Blood-only samples were also imaged as controls for SO₂ unmixing. The spectral data from each acquisition was plotted and compared to its theoretical spectrum, which was calculated as the superposition of optical absorption spectra of 0.3/0.9 probe OD and 50%/70% blood SO₂; agreement between experimental and theoretical spectra was assessed with a correlation coefficient. Note that 1.7 μM PAtrace (i.e., ICG dye concentration in PAtrace formulation) and 5.1 μM ICG (i.e., triple the concentration) are both equal to 0.9 OD when measured at their respective spectral peaks.

PA images were linearly unmixed in MATLAB for HbO₂, HHb, and PAtrace (or ICG). The following wavelengths were used: 710, 734, 760, 800, 830, and 870 nm. The voxel with the maximum 800 nm PA signal was automatically selected on each axial image, a $3 \times 3$ ROI was centered on that voxel, and the SO₂ estimate and PAtrace (or ICG) signal within all ROIs were assessed across all z-slices (see "PA post-processing and ROI selection" section). To validate the temporal stability of blood samples in the sealed PE tubes, two samples (i.e., 50% and 75% SO₂) were tested by injecting the sample into the tube, then monitoring with a repeated, multi-wavelength PA imaging sequence for 5 min to ensure that SO₂ temporal

stability was achieved through the duration of the phantom experiments. In both samples, the $SO_2$ estimates did not change by more than 5%.

The wavelength combination used for unmixing was determined by a cost function, which included: mean $SO_2$ error between PA-based $SO_2$ estimates without and with probe; error in the PAtrace (or ICG) signal ratio from a threefold increase in concentration; and number of wavelengths (Supplementary Fig. 7). The phantom-experiment cost function relies on the fact that we obtain PA-based $SO_2$ estimates of blood samples without exogenous probe (i.e., either PAtrace or ICG) and that we also obtain PA-based probe estimates of blood samples mixed with known probe-concentration increases (i.e., from 0.3 to 0.9 OD). The cost function optimizes for both the accuracy of probe unmixing (i.e., the error between known and PA-based estimates of increases in probe concentration) and $SO_2$ estimation (i.e., the error between PA-based $SO_2$ estimates for a blood sample with and without probe present), while it requires that the same wavelengths be used for both PAtrace and ICG acquisitions to help ensure a fair comparison. To promote faster acquisition times, a modest penalty (i.e., square root) was included for the number of unmixing wavelengths used.

**In vivo comparison of PAtrace to monomeric ICG.** For all in vivo experiments, animal protocols were approved by the Institutional Animal Care and Use Committee of MD Anderson Cancer Center and complied with all relevant ethical regulations for animal testing and research. For a matched comparison of PAtrace and monomeric ICG in vivo, 100 μL of 0.4 mM ICG was injected intravenously into a nu/nu mouse, and PA imaging was conducted on the inVision system (710, 734, 760, 800, 830, 870, 890, 900, and 920 nm; 10-frame averaging) in a region where the aorta and a superficial artery are clearly visible. For all inVision imaging, mice were allowed to equilibrate in the system for 10 min before imaging. Isoflurane was maintained at 2% with an oxygen flow rate of 1 L/min for all mouse imaging. Once the probe completely cleared circulation (~90 min), 100 μL of PAtrace was injected at a matched ICG concentration of 0.4 mM and imaged with the same parameters and in the same anatomical location. All PA acquisitions were unmixed for $HbO_2$, HHb, and PAtrace (or ICG); display dynamic ranges were set to ensure that no preinjection signal from either probe was observed and that comparable postinjection intensity of a superficial artery was achieved.

In a separate experiment, 50 μL each of PAtrace and ICG were injected at matched ICG dye concentrations of 0.01 mM subcutaneously into the mammary fat pad of the same mouse, and a tissue-mimicking stand-off was placed on top of the mouse to achieve a ~17 mm imaging depth for both injected probe samples. PA imaging was acquired for both probes simultaneously with the Acuity system (680–980 nm, 5 nm step size). PAtrace (or ICG) was visualized at its peak wavelength (i.e., 885 nm for PAtrace or 800 nm for ICG) and displayed over the background B-mode US image; display dynamic ranges were set to ensure that no preinjection signal from either probe was observed.

**Toxicity study.** A dose-response assay with PEGylated PAtrace (PEG-PAtrace) was conducted to assess cell viability. PEG-PAtrace and monomeric ICG solutions were prepared in fresh cell culture media with concentrations ranging from 0 to 6.6 OD, where the latter corresponded to ~$5.9 \times 10^9$ PAtrace nanoparticles/mL. Solutions with empty liposomes were similarly prepared with concentrations ranging from 0.34x to 2.7x of the highest PAtrace concentration (i.e., ~$1.57 \times 10^{10}$ nanoparticles/mL, which corresponds to 6.6 OD). The empty liposomes were prepared using the same lipid ratios and concentrations as used in PAtrace synthesis (see "Composition and synthesis of PAtrace" section). SKOV3, A2780, MDA-MB-468, FaDu, 3T3 (ATCC, Manassas, VA), and HUVEC (Thermo Fisher Scientific Cat# C0035C) cells were seeded on 96-well tissue culture polystyrene plates at 8000 cells/well in triplicate of each concentration and incubated overnight at 37 °C in 5% $CO_2$. Then, culture media was replaced with solutions of PEG-PAtrace for all cells, and with ICG or liposomes for HUVEC cells; cells were incubated for another 24 or 48 h. After incubation, culture media with PAtrace was removed, and a mixture of 1-part CellTiter 96 Aqueous Nonradioactive MTS (Promega Corporation, Madison, WI) and 5 parts fresh media was added at 120 μL/well. After incubation for 90 min at 37 °C in 5% $CO_2$, absorbance was measured at 490 nm in a Synergy HT Microplate Reader using Gen5 software. Statistical differences between each sample and the control (i.e., no nanoparticles or ICG) were determined by a two-tailed two-sample $t$-test ($\alpha = 0.05$).

**Animal model of ovarian cancer.** To evaluate the utility of PAtrace for molecularly specific PA imaging in vivo, an orthotopic model of ovarian cancer was used. $FR\alpha^+$ SKOV3 cells were grown and inoculated into mice, as previously described[64,65]. Female athymic nu/nu mice were injected with $1 \times 10^6$ cells into the left ovary via intraovarian surgical implantation. Tumors were allowed to establish over a period of ~14 days.

**In vivo biodistribution imaging protocol.** To monitor SKOV3 tumor growth, axial and coronal T2-weighted MR images were acquired on a 7T Bruker BioSpec (Bruker, Billerica, MA). When tumors reached ~5 mm diameter, 3D tomographic PA imaging was carried out using the inVision system (730, 760, 780, 800, 830, 890, and 910 nm; 15-frame averaging; see "In vivo comparison of PAtrace to monomeric ICG" for other imaging parameters). PA images were acquired axially over a range of z-positions that spanned the liver and intraovarian tumor in 0.5 mm steps, ensuring that the entire tumor volume and multiple control organs (i.e., liver, spleen, and kidneys) were captured. Baseline 3D PA imaging was performed prior to a 100 μL intravenous injection at 100 OD of either $FR\alpha$-PAtrace ($N = 3$ mice) or non-targeted RG-16-PAtrace ($N = 3$ mice). Mice were then imaged immediately, 30 min, and 60 min postinjection. As this is a pilot study, the sample size per group ($N = 3$) was selected as the minimum where we could reasonably obtain estimates of the mean and variance within each group. These estimates will be used to inform the design of subsequent follow-up studies.

**PA post-processing and ROI selection.** For all imaging experiments, "$SO_2$ image" is defined as an image matrix containing the voxel-wise ratio of the unmixed $HbO_2$ coefficient to the total hemoglobin (i.e., $HbO_2 + HHb$)[66], while "$SO_2$ estimate" is a scalar obtained by taking the spatial average of $SO_2$ image over a given ROI. Similarly, "PAtrace (or ICG) image" is an image matrix containing the voxel-wise ratio of unmixed PAtrace (or ICG) coefficient to the sum of the coefficients for all three chromophores (i.e., $HbO_2$, HHb, and PAtrace [or ICG]), while "PAtrace (or ICG) signal" is a scalar obtained by taking the spatial average of PAtrace (or ICG) image over a given ROI.

To improve the accuracy of PA quantification, a wavelength-dependent, surface-fluence-correction model was implemented[16]. The inVision's irradiation field was modeled from the ring geometry, including the wavelength-dependent water absorption, modeled as exponential attenuation over depth from water absorption[67], and approximate beam divergence, modeled as a superposition of Green's functions for the photon diffusion equation (Supplementary Fig. 14). The mouse surface was manually segmented on each of the axial PA images, and surface-fluence correction was applied at each point along the surface mask. A finite element method mesh (~8500 triangular elements) was then generated using NIRFAST 9.1 software[68,69] to smoothly propagate the surface-fluence correction through the volume. The mesh used average tissue parameters ($\mu_a = 10$ m$^{-1}$, $\mu_s' = 100$ m$^{-1}$) based on literature values[70], which were set to be constant across wavelength. After applying surface-fluence correction, PA images were unmixed via linear regression for $HbO_2$, HHb, and PAtrace using all imaging wavelengths (see "In vivo biodistribution imaging protocol" section), as determined by a cost function, which included change in $SO_2$ estimates from preinjection to postinjection, preinjection PAtrace signal, and number of wavelengths (Supplementary Fig. 15). The cost function sought to minimize the preinjection PAtrace signal (which is expected to be zero in the case of perfect unmixing) and minimize the change in the $SO_2$ estimates from the preinjection acquisition to the (immediately) postinjection one in a superficial, $3 \times 3$ ROI in the liver. The latter optimization component is based on the assumption that $SO_2$ in the liver is not expected to change significantly in the ~14 min window immediately prior to and following the PAtrace injection, as blood $SO_2$ would quickly reach equilibrium (within a minute) following the injection. Thus, significant changes in such pre-to-post $SO_2$ estimates are instead assumed a result of unmixing inaccuracies caused by the newly introduced PAtrace chromophore.

"PAtrace CE image" for a particular time-point is defined as

$$\text{PAtrace CE image} = \frac{\text{PAtrace image}_{\text{time-point}} - \text{PAtrace image}_{\text{preinjection}}}{\text{PAtrace image}_{\text{preinjection}}}.$$

This metric signifies that any positive PAtrace CE image voxel has a greater PAtrace signal than the preinjection value, demonstrating the presence of PAtrace in that voxel at that time-point. Note that "PAtrace CE" is a scalar obtained by taking the spatial average of PAtrace CE image over a given ROI. For quantitative comparison of the images, ROIs were manually placed on images encompassing the liver, spleen, and kidneys (i.e., as control organs; 7, 5, and 5 slices each, respectively) and the tumor (5 slices) for each mouse. Tumors were not readily identifiable in PA images, so the tumor position and size were determined by co-registering the MRI-established location and size with the 800 nm PA imaging volume using the left kidney as a fiducial, which was clearly visible in both PA and MRI data. Because PAtrace accumulation was expected to be sparse and focal in the tumor (due to inhomogeneities within the tumor mass), only PAtrace CE image voxels that exceeded the baseline (i.e., noise) variation were included in analysis. This baseline variation was determined by calculating the voxel-wise difference between spatially adjacent preinjection PAtrace images, then taking the mean and SD of these differences within the tumor ROI. For analysis, PAtrace CE image values were excluded if they fell within the mean (±SD) of the baseline variation for each mouse. Finally, $SO_2$ estimates and PAtrace CE were determined in each segmented organ or tumor volume (e.g., through volumetric tumor ROI) for each mouse at each time-point, as shown in Fig. 6. Statistical differences between groups were determined by a one-tailed two-sample $t$-test ($\alpha = 0.05$).

**In vivo circulation half-life imaging and analysis protocol.** To estimate the half-life of $FR\alpha$-PAtrace in circulation, PA imaging was carried out in the inVision system (890 nm; 15-frame averaging). PA images were acquired axially over three z-positions (0.5 mm step size) spanning the neck. Baseline 3D PA imaging was performed on wild-type female athymic nu/nu mice before 100 μL intravenous injection of 100 OD $FR\alpha$-PAtrace, then mice were imaged continuously for 15 min immediately post injection. For analysis, an ROI was manually placed over a superficial surface vein. Analysis was done from the center of the vessel, which was

determined to be the voxel with the maximum sum across time. This voxel was plotted across time and the data were fit to an exponential as

$$PA(t) = PA_0 e^{-\lambda t},$$

where $PA_0$ is the maximum 890 nm PA signal, $t$ [min] is the time from $PA_0$, and $\lambda$ [min$^{-1}$] is the decay constant. The half-life was then calculated as $t_{1/2} = \ln(2)/\lambda$.

**Blood assays and histology**. To determine in vivo biosafety of FRα-PAtrace, hematology, and blood chemistry analysis and histology of vital organs were performed on wild-type nu/nu mice injected with FRα-PAtrace ($N = 5$) sacrificed 24 h postinjection and on one control mouse that was not injected. Blood samples were collected via cardiac puncture, and samples were analyzed on a Siemens ADVIA 120 Hematology System (Siemens Medical Solutions USA, Inc., Malvern, PA) and a Roche COBAS Integra 400 Plus (Roche Diagnostics, Rotkreuz, CH). Blood assays were evaluated by a veterinarian certified by the American College of Laboratory Animal Medicine. For histology, the liver, spleen, kidneys, and heart were collected immediately after sacrifice, and tissues were fixed in formalin for 48 h before processing with H&E staining. Tissues were evaluated by a veterinary pathologist certified by the American College of Veterinary Pathologists.

**Western blot of SKOV3 tumors**. To prepare lysates of snap-frozen tissue from mice, approximately 30 mm$^2$ cuts of tumor tissue were disrupted with a tissue homogenizer and subjected to centrifugation at 20,784$g$ for 30 min in modified RIPA buffer. The protein concentrations were determined using a BCA Protein Assay Reagent kit (Pierce Biotechnology, Rockford, IL). Lysates were loaded and separated by Nu-PAGE 4–12% gradient gels. Proteins were transferred to a nitrocellulose membrane by iBlot (Thermo Fisher Scientific), blocked with 4% bovine serum albumin, and washed with a mixture of tris-buffered saline and Tween 20 solutions. Membranes were then incubated at 4 °C with primary anti-FRα monoclonal antibodies (2 µg/mL; clone 548908, Thermo Fisher Scientific Cat# MA5-23917) overnight. After washing, membranes were incubated with horseradish peroxidase (HRP)-conjugated sheep secondary anti-mouse IgG (1:3000; Thermo Fisher Scientific Cat# 45000679) for 2 h. HRP was visualized by an enhanced chemiluminescence detection kit (Thermo Fisher Scientific). Blots were re-probed with an anti-β-actin antibody (0.1 µg/mL; clone AC-15, Sigma-Aldrich Cat# A5441) as a loading control.

**Reporting summary**. Further information on research design is available in the Nature Research Reporting Summary linked to this article.

## Data availability

All data supporting the findings from this study are provided in the paper and its supplementary information. TEM images are provided at: https://www.ebi.ac.uk/biostudies/preview/BioImages/studies/S-BIAD167. Source data are provided with this paper. Any remaining raw data will be available from the corresponding authors upon reasonable request. Source data are provided with this paper.

## Code availability

The custom MATLAB code used in this study is available at: https://github.com/RichardBouchard/PAtrace_NatCommun.

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

## Acknowledgements

This work was supported by the Cancer Prevention & Research Institute of Texas [RP170314 and RP190131], the National Institutes of Health [R01 EB028762; P30 CA016672; P50 CA217685; and S10 OD019946], the American Cancer Society, the Frank McGraw Memorial Chair in Cancer Research, and the Sao Paulo Research Foundation [2016/22720-3]. We would like to thank Charles Kingsley and Caterina Kaffes for help with animal preparation and imaging; Trevor Mitcham for assisting in phantom preparation; Kenneth Dunner for assistance with TEM imaging; Kelly Kage, MFI, CMI, for her assistance with medical graphics; and Dr. Elizabeth Whitley, DVM, PhD, DACVP for evaluation of histology slides.

## Author contributions

K.V.S. conceived the project; K.V.S. and R.R.B. designed the experiments, analyzed the data, and interpreted the results; J.T.H., K.A.H., J.R.C., S.Y.E., and K.V.S. developed the nanoparticle formulation and synthesis process; S.H., C.S.K., and J.R.C. performed particle characterization experiments; S.H., D.R.T.S., and R.R.B. designed PA phantoms and performed phantom imaging; S.H. performed in vitro stability and toxicity experiments; J.L.S. reviewed the blood assays; Y.W. and A.K.S. developed the ovarian cancer mouse model; Y.W., A.K.S., K.V.S., and R.R.B. designed animal imaging studies; C.A.W., C.S.K., and R.R.B. performed in vivo PA imaging experiments; C.A.W. and R.R.B. developed algorithms for quantitative PA image analysis; C.A.W., S.H., C.S.K., Y.W., J.C., K.V.S., and R.R.B. co-wrote the paper; all authors contributed to data analysis, discussion of results, and paper editing.

## Competing interests

K.A.H. is a consultant for Cellino Bio and is a shareholder in NanoHybrids, Inc. and Senda Biosciences, Inc. A.K.S. is a consultant for Merck and Kiyatec, is a shareholder in BioPath, and receives research support from M-Trap. K.V.S., J.T.H., and J.R.C. are shareholders in NanoHybrids, Inc. C.A.W., S.H., C.S.K., Y.W., D.R.T.S., J.L.S., S.Y.E., and R.R.B. declare no competing interests.
