## [Peer Review File · Nature Communications]

REVIEWER COMMENTS

Reviewer #1 (Remarks to the Author):

Manuscript ID: NCOMMS-20-20003

Manuscript Title: Clinically translatable quantitative molecular photoacoustic imaging with liposome-encapsulated ICG J-aggregates

This manuscript reported a novel ICG J-aggregation loaded nanoparticles (PAtrace), which is very promising to address the clinical requirement for a photoacoustic contrast. As-prepared PAtrace show a markedly red-shifted sharp absorption peak at 890 nm in comparison with monomeric ICG. Such sharp absorption peak lies in a spectral region where hemoglobin absorption spectra are relatively flat, allowing straightforward unmixing of hemoglobin and PAtrace with high in vivo contrast.

In my opinion, the authors were able to come up with a very exciting strategy to achieve potential clinically translatable PA contrast. The concept is very interesting. The authors put in tremendous effort in well planned out and executed experiments to support their findings. The data is solid. The discussions are reasonable, and the conclusions are sound. Thus, this work could be published in nature communications after minor revision.

1. Detailed preparation of PAtrace as well as the characterization of the FR α -PAtrace should be provided.
2. Thermal stability of PAtrace is of great importance for high-performance PA imaging, is there any chance of disaggregation for ICG J-aggregation at considerably high temperature. As known, PA imaging usually lead to local hyperthermia.
3. How about the fluorescence of PAtrace, red-shift into second window (1000-1700 nm)? If so, could PAtrace to be used as fluorescence contrast for second window fluorescence imaging?
4. The scale bar for the in vivo PA imaging should be given (for example, Fig. 5b).
5. The title of the reference has different formats, especially the lowercase and the uppercase.

Reviewer #2 (Remarks to the Author):

This paper describes a nanoparticle based agent for photoacoustics. It is (mostly) well written easy to follow. The demonstrated nanoparticle seems to be working well, is optimized for MSOT and improves over ICG alone. The additional targeting factor has a clear advantage over non targeted (EPR) retention.

However, some issues remain:

1. How the J-aggregates are prepared, method should be provided.
2. How J-aggregates are loaded into liposomes, mention in the methods. W
3. hat is the loading efficiency of J-aggregates, a comparison with conventional ICG dye should be included.
4. For better understanding show the normalized figure 2d.
5. For better understanding show the image of the phantom and the ROI for figure 2d and e.
6. Optical density has unit, the label of x-axis of figure 2e is not correct.
7. For better understanding OD and Scattering coefficient of the samples in Supplementary Figure 4 should be mentioned.
8. Cancer cell lines may not be the best cell line for toxicity test as they are known to be very robust. A non-cancerous cell type (e.g., endothelial cells) would be a better model to assess cell toxicity of PAtrace.
9. Although mentioned in the methods, the distribution imaging data in all vital organs in not

provided, data for spleen and kidney along with liver and tumor should be included.

10. For clinical translation, in vivo biosafety of PAtrace is essential. Please provide blood chemistry parameter and hematology analysis and histology of vital organs.

11. Statement on sample size (animal number) justification should be provided.

Reviewer #3 (Remarks to the Author):

The paper describes the development of a novel photoacoustic contrast agent based on liposomes containing J-aggregates of indocyanine green bioconjugated to target the FR α receptor in an ovarian cancer model. The photoacoustic performance of the probe is characterized in phantoms, its toxicity is characterized in cell cultures and the targeting efficacy in tumor-bearing mice. The authors should consider the following comments.

It is not clear what is the novelty of the paper. In the methods section, it is mentioned that the PAtrace probe is commercially available. Thereby, the claims related to the development of a probe featuring i) strong absorbance at $\sim 890\text{nm}$, ii) PA signal enhancement due to J-aggregation and iii) concentration-independent spectrum are not really part of this paper. Only claim iv) molecular specificity in the abstract can be considered novel. Functionalization of the probe to target tumor receptors is indeed interesting and challenging. However, several parts of the text read like the PAtrace probe has been synthesized in this work, e.g. "we developed and validated a PA contrast agent, PAtrace" (introduction). I recommend that the claims of the paper are better defined.

It is also highlighted throughout the text that the probe is clinically translatable (or potentially clinically translatable). This is also the main selling point in the title. The main argument for asserting this is that the components of the nanoparticles (ICG, liposomes) are FDA approved, which can facilitate the regulatory process. However, it is not clear that that is the case. Different types of ICG nanoparticles, including liposomes loaded with ICG have been reported and none of them is close to be FDA approved. Also, to the best of my knowledge, no targeted optical contrast agent is FDA approved, which may also represent a burden for the suggested targeted nanoparticles. I would be more moderate in this claim.

One of the limitations of ICG (mentioned in the introduction) is its short half-life circulation, which limits its use in disease monitoring. I miss a proper characterization of the half-life circulation of the developed particles. This should be possible to do considering that real-time photoacoustic imaging systems are available to the authors.

How did the authors make sure that the dye molar concentration of ICG in its monomeric form and in the nanoparticle encapsulation is the same in the characterization experiments shown in Fig. 2?

I would recommend that the results shown in Figs. 2d and 2e are compared to the equivalent ones obtained with monomeric ICG. From the literature, it appears that the spectrum of ICG in plasma is stable at least up to $65\mu\text{M}$ ($\sim 13\text{OD}$). Also, from Suppl. Fig. 4, it appears that the linearity of the signal with fluence is higher for monomeric ICG.

I am wondering how was ensured that the oxygenation in the blood used for the phantoms was maintained at 50% and 70%. It may change e.g. due to contact with air. Generally, very sophisticated systems have been used to achieve a constant oxygenation in a tubing. The authors refer to previous work for this in the methods, but I think it is important to describe it here.

It is said that the unmixing in phantoms was done with wavelength combinations chosen to best fit ICG and PAtrace. This appears to be arbitrary and I don't think it represents a fair comparison. I would recommend to use the entire wavelength range in both cases, particularly since this is also required for accurate sO_2 estimations.

What is the reason for a relatively low ICG signal increase of 1.4 for 3-fold increase in

concentration in Fig. 3d. As I mentioned above, the monomeric ICG signal as a function of concentration could as well be measured.

The authors highlight that the phantoms used for the results shown in Figs. 3 and 4 are "tissue-mimicking". It is not clear for me that is the purpose of this. For example, in the cell culture experiments it does not seem to be important the phantom composition. In the experiments shown in Fig. 3, the phantom contains ink as background absorption, which does not reflect the spectral-coloring of light at deep locations. If this is not the purpose, one may just use a transparent phantom featuring no light attenuation. What type of gelatin was used for building the phantoms?

I am puzzled about the result in Fig. 3c. Why would the error in sO₂ change with the concentration of the probe if the signal is linear with fluence and it is measured at the same point (no differences in spectral coloring)?

The authors mention that cytotoxicity of PAttrace at high concentrations may be associated to the lipid coating. This should be assessed e.g. by considering only liposomes or PAttrace probe with no targeting moiety.

I would recommend to make experiments that enable characterizing the extravasation of PAttrace and how this compares with that of targeted PAttrace. For example, in Ermolayev et al. *European Radiology* 26 (6), 1843-1851 (2016), the authors use a real-time imaging system to visualize the extravasation of liposomal ICG. I think this is very interesting and doable with the real-time imaging system available to the authors.

I am also missing a characterization of the photo-stability of the PAttrace with and without targeting moiety and how this compares with monomeric ICG.

AUTHOR'S RESPONSE TO EDITORS

Reviewer #1 (Remarks to the Author):

This manuscript reported a novel ICG J-aggregation loaded nanoparticles (PAtrace), which is very promising to address the clinical requirement for a photoacoustic contrast. As-prepared PAtrace show a markedly red-shifted sharp absorption peak at 890 nm in comparison with monomeric ICG. Such sharp absorption peak lies in a spectral region where hemoglobin absorption spectra are relatively flat, allowing straightforward unmixing of hemoglobin and PAtrace with high in vivo contrast.

In my opinion, the authors were able to come up with a very exciting strategy to achieve potential clinically translatable PA contrast. The concept is very interesting. The authors put in tremendous effort in well planned out and executed experiments to support their findings. The data is solid. The discussions are reasonable, and the conclusions are sound. Thus, this work could be published in nature communications after minor revision.

1. Detailed preparation of PAtrace as well as the characterization of the FR α -PAtrace should be provided.

Thank you for pointing out this omission. We have added a detailed protocol for synthesis of PAtrace to the “*Composition and synthesis of PAtrace*” section of the Methods. We have also added characterization of FR α -PAtrace UV-Vis-NIR spectra, photoacoustic (PA) spectra, and photothermal stability to the “*In vitro evaluation of FR α -PAtrace*” section of the Results. Briefly, a lipid-thin film was hydrated in a concentrated solution (1 mg/mL) of ICG dye under sonication using a probe sonicator. This induced the formation of liposomes containing concentrated ICG solution. The solution was then left at room temperature for ~60 days of “maturation” in the dark. At this point, J-aggregates of ICG are formed both inside and outside of the liposomes. Lastly, a purification procedure was performed, which removes any free J-aggregates and monomers of ICG from the liposomes containing ICG J-aggregates.

2. Thermal stability of PAtrace is of great importance for high-performance PA imaging, is there any chance of disaggregation for ICG J-aggregation at considerably high temperature. As known, PA imaging usually lead to local hyperthermia.

This is an excellent point, and we apologize for not addressing it in our original submission. We have since assessed the thermal stability of targeted PAtrace (i.e., FR α -PAtrace) to over 1,000 laser pulses near the ANSI fluence limit for skin (i.e., 20 mJ/cm²) to create a “worst-case” heating scenario. As is now shown in Supplementary Fig. 11, targeted PAtrace presents with the same final spectrum – both in shape and in magnitude – as is presented in the initial spectrum, *before* significant laser-pulse irradiation. Additionally, these spectra are very similar to the UV-Vis-NIR spectrum for PAtrace (see Fig. 2c). Supplementary Fig. 4a also demonstrates the stability of PAtrace exposed to fluences in excess of 20 mJ/cm².

3. How about the fluorescence of PAtrace, red-shift into second window (1000-1700 nm)? If so, could PAtrace to be used as fluorescence contrast for second window fluorescence imaging?

This is a very interesting suggestion. Previous studies of fluorescence properties of ICG J-aggregates showed a very low quantum yield of $\sim 3 \times 10^{-4}$, with a fluorescence emission peak that overlapped with the absorbance peak of ICG J-aggregates (Rotermund *et.al.*, *J. of Photochemistry and Photobiology A: Chemistry*, 110(1), 75-78, 1997). Indeed, we have not observed significant fluorescence from PAtrace with excitations up to 950 nm. Currently, we do not have access to an experimental set-up that would allow for exploration of potential fluorescence in the 1000-1700 nm window.

4. The scale bar for the in vivo PA imaging should be given (for example, Fig. 5b).

We apologize for this oversight. The scale bar for the PA images has since been added.

5. The title of the reference has different formats, especially the lowercase and the uppercase.

We apologize for inconsistencies in the references. All references have been checked and revised as needed to ensure correct and consistent formatting.

Reviewer #2 (Remarks to the Author):

This paper describes a nanoparticle based agent for photoacoustics. It is (mostly) well written easy to follow. The demonstrated nanoparticle seems to be working well, is optimized for MSOT and improves over ICG alone. The additional targeting factor has a clear advantage over non targeted (EPR) retention.

However, some issues remain:

1. How the J-aggregates are prepared, method should be provided; and 2. How J-aggregates are loaded into liposomes, mention in the methods.

Thank you for pointing out this omission. We have added a detailed protocol for the synthesis of PAttrace to the “*Composition and synthesis of PAttrace*” section of the Methods to address both of the aforementioned issues. Briefly, a lipid-thin film was hydrated in a concentrated solution (1 mg/mL) of ICG dye under sonication using a probe sonicator. This induced the formation of liposomes containing concentrated ICG solution. The solution was then left at room temperature for ~60 days of “maturation” in the dark. At this point, J-aggregates of ICG are formed both inside and outside of the liposomes. Lastly, a purification procedure was performed, which removes any free J-aggregates and monomers of ICG from the liposomes containing ICG J-aggregates.

3. What is the loading efficiency of J-aggregates, a comparison with conventional ICG dye should be included.

We apologize for not originally including an estimate of J-aggregate loading efficiency. To address this, we have since added quantitation of the loading efficiency of PAttrace as the number of ICG dye molecules per nanoparticle. Details of the procedure used have been added to the newly included “*Composition and synthesis of PAttrace*” section of the Methods. In summary, the concentration of PAttrace nanoparticles was determined by tunable resistive pulse sensing (Weatherall *et al.*, *Analytical Chemistry*, 88(17), 8648-8656, 2016) using a qNano Gold system (Izon Science Ltd., Oxford, UK). To measure the concentration of ICG dye molecules, the PAttrace sample was mixed with ethanol to achieve a 1:1 v/v ethanol/water ratio. Under this condition, the liposomal coating was dissolved, causing ICG J-aggregates to dissociate to monomeric ICG molecules almost instantly. Then, the absorbance of the ICG solution at 790 nm was measured, and the concentration of ICG molecules was determined using a linear calibration curve of ICG ($R^2 = 0.999$) in 1:1 v/v ethanol/water solution. The data show that each PAttrace nanoparticle contains $\sim 1.54 \times 10^6$ ICG molecules. This loading efficiency of ICG dyes per liposomal nanoparticle is $\sim 4 \times 10^2$ greater than what is reported for liposomes loaded with monomeric ICG dyes (Beziere *et al.*, *Biomaterials*, 37, 415-424, 2015). This significant increase in the loading efficiency of PAttrace is most likely associated with a dense packing of ICG dyes in J-aggregates. Comparable increases in loading efficiency were reported for pharmaceutical drugs, such as DOXIL, where a dense packing of doxorubicin molecules into crystals inside liposomes results in an increase in a drug’s loading by >100x compared to dispersed molecules in solution (Barenholz *et al.*, *Journal of Controlled Release*, 160(2), 117-134, 2012). We have since added these loading efficiency data to the “*Characterization of PAttrace*” section in the Results.

4. For better understanding show the normalized figure 2d.

Thank you for this suggestion. As a result, Fig. 2e is now a plot of the normalized PA spectra for all concentrations of PAttrace tested. We agree that presenting this data as such more clearly shows the spectral stability of PAttrace with varied concentration. The plot previously provided as Fig. 2e, which shows PA signal linearity as a function of PAttrace concentration, has now been incorporated in Fig. 2d as a figure inlay.

5. For better understanding show the image of the phantom and the ROI for figure 2d and e.

We apologize for not being clearer about this in the initial submission. The phantom used to acquire the data for Figs. 2d-e is the same phantom configuration depicted in Fig. 4c. This important point has now been made clearer in the text. Schematics and photographs of the phantom mold and an example phantom have also now been provided in the new Supplementary Fig. 5. Additionally, phantom-component labels have been added to Fig. 4c to improve understanding of the phantom’s layout.

6. Optical density has unit, the label of x-axis of figure 2e is not correct.

We apologize for this oversight. The designation “unitless” has been removed from the x-axis label of this figure. Please, note that this figure is now an inlay in Fig. 2d in the revised manuscript

7. For better understanding OD and Scattering coefficient of the samples in Supplementary Figure 4 should be mentioned.

Unfortunately, we currently do not have access to a system that would allow us accurate measurements of the scattering coefficient of the tested samples. However, given the similarity observed between PA spectra – which results from absorption – and the absorbance spectra measured by our spectrophotometer, we believe that absorption is the dominant contributor to the UV-Vis-NIR spectra of the samples. As such, we have added the absorbance spectra of the tested samples at the optical density (OD) that was used during PA measurements as Supplementary Fig. 4b.

8. Cancer cell lines may not be the best cell line for toxicity test as they are known to be very robust. A non-cancerous cell type (e.g., endothelial cells) would be a better model to assess cell toxicity of PAtrace.

We agree with the reviewer. To address this concern, we have added a toxicity assessment of PAtrace in normal endothelial cells, HUVEC, to Supplementary Fig. 12. Furthermore, we have also evaluated cytotoxicity of the two major components of PAtrace – the liposomes and the ICG dye molecules – alone in the HUVEC cell line. These data are now shown in Supplementary Fig. 13 and discussed in the Results. Briefly, PAtrace exhibited a cytotoxic effect in HUVEC cells only at the highest concentration of 6.6 OD. No cytotoxicity was observed for empty liposomes. With ICG alone, HUVEC cells exhibited cytotoxicity at high ICG concentrations of 3.3 and 6.6 OD for both time-points (24 and 48 hours), as shown in Supplementary Fig. 13. These data indicate that the observed minor cytotoxic effect at high PAtrace concentrations might be associated with a very high loading of ICG dye molecules inside PAtrace.

9. Although mentioned in the methods, the distribution imaging data in all vital organs in not provided, data for spleen and kidney along with liver and tumor should be included.

We apologize for this oversight in the initial submission. We have since provided additional analysis for biodistribution of PAtrace in the spleen and kidney. As shown in Fig. 6, the spleen presents with similar PAtrace Contrast Enhancement (CE) to that observed in the liver. Conversely, we observed no PAtrace CE in the kidneys, indicating that PAtrace does not accumulate renally. Both spleen and kidney biodistribution are similar to that observed in the study by Song *et al.* (*RSC Advances*, 5(5), 3807-3813, 2015), which investigated uptake of *monomeric* ICG encapsulated in liposomes using PA imaging and also observed no accumulation in the kidney. As Song *et al.* noted: “However, no obvious ICG accumulation happened in kidney for ICG or Lipo-ICG, indicating that neither of these two formulations are cleared renally. The observations in spleen were similar to those in liver, with free ICG quickly diffusing into and out of the spleen, while Lipo-ICG gradually accumulated there.”

10. For clinical translation, in vivo biosafety of PAtrace is essential. Please provide blood chemistry parameter and hematology analysis and histology of vital organs.

Thank you for this insightful suggestion. Hematology (Supplementary Table 1), blood chemistry (Supplementary Table 2), and histology (Supplementary Fig. 18) results for non-tumor-bearing mice (N=5) injected with FR α -PAtrace and one control mouse are now provided. A summary of these results prepared by a board-certified veterinarian (i.e., hematology and blood chemistry) and a board-certified veterinary pathologist (i.e., histology) are provided in the newly added “*In vivo biosafety and circulation half-life of PAtrace*” section of the Results. Briefly, hematology and blood chemistry showed no evidence of toxicity as a result of FR α -PAtrace injection, and no morphologic changes were observed on histology of the liver, spleen, kidney, and heart.

11. Statement on sample size (animal number) justification should be provided.

As this is a pilot study, the sample size per group (N=3) was selected as the minimum where we could reasonably obtain estimates of the mean and variance within each group. These estimates will be used to inform the design of subsequent follow-up studies. This has now been clarified in the manuscript.

Reviewer #3 (Remarks to the Author):

The paper describes the development of a novel photoacoustic contrast agent based on liposomes containing J-aggregates of indocyanine green bioconjugated to target the FR α receptor in an ovarian cancer model. The photoacoustic performance of the probe is characterized in phantoms, its toxicity is characterized in cell cultures and the targeting efficacy in tumor-bearing mice. The authors should consider the following comments.

1. It is not clear what is the novelty of the paper. In the methods section, it is mentioned that the PAttrace probe is commercially available. Thereby, the claims related to the development of a probe featuring i) strong absorbance at ~890nm, ii) PA signal enhancement due to J-aggregation and iii) concentration-independent spectrum are not really part of this paper. Only claim iv) molecular specificity in the abstract can be considered novel. Functionalization of the probe to target tumor receptors is indeed interesting and challenging. However, several parts of the text read like the PAttrace probe has been synthesized in this work, e.g. “we developed and validated a PA contrast agent, PAttrace” (introduction). I recommend that the claims of the paper are better defined.

We apologize for the confusion regarding the novelty of our work. We would like to clarify that PAttrace is not commercially available. This contrast agent was initially devised in Dr. Sokolov’s lab, and its synthesis was developed in a collaboration with his lab and NanoHybrids, Inc. This paper is the first report with a thorough characterization of this contrast agent, and PAttrace was fully developed in the work presented in this manuscript. Therefore, all of our claims regarding the PAttrace agent are novel as they have not been reported previously in a peer-reviewed publication.

2. It is also highlighted throughout the text that the probe is clinically translatable (or potentially clinically translatable). This is also the main selling point in the title. The main argument for asserting this is that the components of the nanoparticles (ICG, liposomes) are FDA approved, which can facilitate the regulatory process. However, it is not clear that that is the case. Different types of ICG nanoparticles, including liposomes loaded with ICG have been reported and none of them is close to be FDA approved. Also, to the best of my knowledge, no targeted optical contrast agent is FDA approved, which may also represent a burden for the suggested targeted nanoparticles. I would be more moderate in this claim.

We agree. In order to fully clarify our claim of translatability, we have added a detailed description after the third paragraph of the Discussion that provides the reasoning behind our expectation that PAttrace’s composition of FDA-approved components would make it a promising agent for future clinical translation. This section reads as follows:

“We carefully selected the composition of PAttrace to simplify future clinical translation. Every major component in the core formulation (i.e., DPPC, DSPE-PEG, DSPE-PEG-Maleimide, cholesterol, ICG, & anti-FR α antibody) can be sourced from sources compliant with current Good Manufacturing Practices (cGMP), which is one major hurdle in clinical translation of a product. Another major barrier to the clinic is the toxicity profiling of each of the components. If a component is not classified as Generally Regarded as Safe or has not had toxicology or pharmacology studies under current Good Laboratory Practices (cGLP), then an extensive amount of preclinical testing is required before clinical testing can occur⁴⁷. Each of PAttrace’s components either have been already used in a clinical trial or is currently approved by the FDA. DPPC has made it to Phase III clinical trials in the formulations Stimuvax (Oncothyreon, Inc., R Flourney, CA) and ThermoDox (Celsion Corporation, Lawrenceville, NJ)⁴⁸. DSPE-PEG has been featured in both FDA-approved Doxil® (Sequus Pharmaceuticals, Inc., Menlo Park, CA) and Onivyde™ (Merrimack Pharmaceuticals, Cambridge, MA)⁴⁸. DSPE-PEG-Maleimide is currently featured in a Phase I clinical trial of MCC-465 (Mitsubishi Tanabe Pharma Corporation, Osaka, JP)⁴⁸. Cholesterol is included in the majority of clinical liposome agents, including Doxil®, DaunoXome® (NeXstar Pharmaceuticals, Boulder, CO), and others⁴⁸. ICG has been FDA approved since 1959⁴⁹. Finally, anti-FR α antibodies have been studied in Phase III clinical trials under the name farletuzumab (Morphotek, Inc., Exton, PA)⁵⁰. Overall, given the clinical history of these components and their relatively low toxicities, we would expect a streamlined path for PAttrace through cGMP, cGLP, and a Phase I clinical trial.”

3. One of the limitations of ICG (mentioned in the introduction) is its short half-life circulation, which limits its use in disease monitoring. I miss a proper characterization of the half-life circulation of the developed particles. This should be possible to do considering that real-time photoacoustic imaging systems are available to the authors.

We apologize for not including such characterization in our initial submission. To address this, we have since performed an experiment to characterize the circulation half-life of PAttrace, with our methodology now provided in a newly included “In

vivo circulation half-life imaging and analysis protocol” section in the Methods and a summary of the results provided in the “*In vivo biosafety and circulation half-life of PAttrace*” section of the Results. Additionally, half-life data plots are now provided as Supplementary Fig. 17. Based on this additional experiment, the half-life of PAttrace was measured to be an average (N=4 mice) of 10 min, which is consistent with the half-life reported by Song *et al.* (*RSC Advances*, 5(5), 3807-3813, 2015) for non-J-aggregated ICG encapsulated in a liposome.

4. How did the authors make sure that the dye molar concentration of ICG in its monomeric form and in the nanoparticle encapsulation is the same in the characterization experiments shown in Fig. 2?

We apologize that this was not clear in the manuscript. The spectral comparison shown in Fig. 2c was obtained from the same sample of PAttrace before (green trace) and after (gold trace) rupturing the liposomal shell with Tween 20. Note that Tween 20 dissociates ICG J-aggregates. Also, at the concentration used in our experiment, Tween 20 does not appreciably change the absorbance spectrum of ICG. These experimental details are described in the “*Spectral comparison of PAttrace to monomeric ICG*” section of the Methods. We have also clarified how this experiment was conducted in the first paragraph of the Results.

5. I would recommend that the results shown in Figs. 2d and 2e are compared to the equivalent ones obtained with monomeric ICG. From the literature, it appears that the spectrum of ICG in plasma is stable at least up to 65 μ M (~13OD). Also, from Suppl. Fig. 4, it appears that the linearity of the signal with fluence is higher for monomeric ICG.

We appreciate the reviewer’s concerns regarding comparisons of linearity with OD between PAttrace and ICG. To address this, we performed the same linearity experiment with monomeric ICG in bovine serum albumin and included the data as Supplementary Fig. 6. We do observe linearity in the ICG spectra up to 12.5 OD; however, at 25 OD, the spectrum is no longer linear, whereas it is for PAttrace (Fig. 2d). At higher concentrations (e.g., 25 OD), monomeric ICG starts forming J-aggregates, which cause a shift in the spectral peak away from 810 nm (i.e., where it was imaged for linearity assessment). Note that the absorption peak of monomeric ICG shifts from 790 nm to 810 nm when it binds to albumin, which is present in blood serum (Landsman *et al.*, *Journal of Applied Physiology*, 40(4), 575-583, 1976).

6. I am wondering how was ensured that the oxygenation in the blood used for the phantoms was maintained at 50% and 70%. It may change e.g. due to contact with air. Generally, very sophisticated systems have been used to achieve a constant oxygenation in a tubing. The authors refer to previous work for this in the methods, but I think it is important to describe it here.

We apologize for not including these details in the manuscript. After injecting samples into the polyethylene (PE) tubes, the ends were sealed to ensure that no changes could occur due to contact with the environment. To validate the temporal stability of oxygenated samples in the sealed PE tubes, a phantom was prepared, as described in the “*Tissue-mimicking phantom comparison of PAttrace to monomeric ICG*” section of the Methods for longitudinal SO₂ monitoring with PA imaging. In short, a CO-oximeter was used to confirm the initial sample SO₂, and two samples (i.e., 50% & 75% SO₂) were injected in different PE tubes, where they were monitored with a repeated PA imaging sequence to ensure that SO₂ temporal stability was achieved through the duration of the aforementioned phantom experiments. Over 5 min, the measured values did not change by more than 5% SO₂ in either sample. The imaging duration for all tissue-mimicking phantom experiments was ~2 min; thus, there was sufficient SO₂ stability for all phantom imaging.

7. It is said that the unmixing in phantoms was done with wavelength combinations chosen to best fit ICG and PAttrace. This appears to be arbitrary and I don’t think it represents a fair comparison. I would recommend to use the entire wavelength range in both cases, particularly since this is also required for accurate sO₂ estimations.

Thank you for pointing out the discrepancy in the wavelengths reported for the tissue-mimicking phantom study. Initially, we investigated the unmixing performance of the wavelengths listed in the paper, which were chosen based on absorption spectral features. However, the wavelengths used for the presented results were ultimately determined based on cost functions, which have now been added as Supplementary Fig. 7 for the aforementioned phantom experiment and as Supplementary Fig. 15 for the *in vivo* biodistribution experiment. The phantom-experiment cost function relies on the fact that we obtain PA-based SO₂ estimates of blood samples without exogenous probe (i.e., either PAttrace or ICG) and that we also obtain PA-based probe estimates of blood samples mixed with known probe-concentration increases (i.e., from 0.3 to 0.9 OD). Similar to the reviewer suggestion, this cost function combines both the accuracy of probe unmixing (i.e., error

between known and PA-based estimates of increases in probe concentration) and SO₂ estimation (i.e., error between PA-based SO₂ estimates for a blood sample with and without probe present), while it requires that the same wavelengths be used for both PAttrace and ICG acquisitions to help ensure a fair comparison. Thus, the phantom results included in our initial submission were actually based on unmixing from just one set of wavelengths (i.e., 710, 734, 760, 800, 830, and 870 nm), which were determined with the cost-function analysis included in Supplementary Fig. 7. Consequently, the wavelengths (i.e., 760, 800, 830, 870, and 890 nm) for PAttrace unmixing initially provided in the “*Tissue-mimicking phantom comparison of PAttrace to monomeric ICG*” section of the Methods were incorrectly reported. We have now corrected this in the manuscript resubmission. We have also (re)checked the imaging and post-processing parameters for all other studies to ensure that everything is correctly reported. We apologize for this initial oversight.

The cost function used to obtain the unmixing wavelengths for the *in vivo* biodistribution experiment (see Supplementary Fig. 15) sought to minimize the preinjection PAttrace signal (which is expected to be zero in the case of perfect unmixing) and minimize the change in SO₂ estimates from the preinjection acquisition to the (immediately) postinjection one within a superficial region of interest (ROI) in the liver. The latter optimization component assumes that SO₂ in the liver does not change significantly in the ~14-min window immediately prior to and following the PAttrace injection, as blood SO₂ would quickly reach equilibrium (within a minute) following the injection. Thus, significant changes in such pre-to-post SO₂ estimates are instead assumed to be a result of unmixing inaccuracies caused by the newly introduced PAttrace chromophore. To promote faster acquisition times, both cost functions include a modest penalty (i.e., square root) for the number of unmixing wavelengths used.

8. What is the reason for a relatively low ICG signal increase of 1.4 for 3-fold increase in concentration in Fig. 3d. As I mentioned above, the monomeric ICG signal as a function of concentration could as well be measured.

Thank you for the suggestion. As a result, we have now included a plot of monomeric ICG PA signal linearity as a function of concentration in Supplementary Fig. 6c. As is shown in this plot, the PA signal for monomeric ICG also remains quite linear for concentrations <15 OD. At higher concentrations (e.g., 25 OD), it starts forming J-aggregates, which cause a shift in the spectral peak. The less-than-expected unmixed ICG signal increase is not due to nonlinearity of the monomeric ICG PA signal, which Supplementary Fig. 6c shows to be quite linear at the concentrations used, but rather due to the confounding hemoglobin spectra that are also included while unmixing. This point has since been clarified in the manuscript. Because the absorption peak for monomeric ICG is within the wavelength range (i.e., ~750-850 nm) most sensitive for unmixing oxy- and deoxyhemoglobin (HbO₂ & HHb), it is difficult to spectrally differentiate ICG from hemoglobin. Thus, some of the increase in PA signal due to an increase in monomeric ICG concentration gets inaccurately assigned to hemoglobin during spectral unmixing. Given the significant underestimation (>30%) of SO₂ observed with increasing concentrations of monomeric ICG depicted in Fig. 3c, it appears much of this increased signal is falsely ascribed to HHb.

9. The authors highlight that the phantoms used for the results shown in Figs. 3 and 4 are “tissue-mimicking”. It is not clear for me that is the purpose of this. For example, in the cell culture experiments it does not seem to be important the phantom composition. In the experiments shown in Fig. 3, the phantom contains ink as background absorption, which does not reflect the spectral-coloring of light at deep locations. If this is not the purpose, one may just use a transparent phantom featuring no light attenuation. What type of gelatin was used for building the phantoms?

For the experiments in Fig. 3, we were interested in how PA-based estimates of agent concentration and SO₂ would be affected at relevant agent concentrations and SO₂ levels but with SNR that would begin to emulate the *in vivo* condition at depth. Thus, standard optical scatterers (i.e., Intralipid) and absorbers (i.e., India ink) were added to the phantom background at concentrations consistent with validated tissue-mimicking levels established in our previous work (Cook *et al.*, *Biomedical Optics Express*, 2(11), 3193-3206, 2011). Conversely, the gelatin-well phantom presented in Fig. 4 was not intended to be tissue-mimicking as it was used merely to immobilize the cells in each well for PA imaging and not to achieve relevant SNR levels at depth. This point has now been clarified in the manuscript. In both phantoms, we used 225-Bloom, Type B gelatin from MilliporeSigma.

10. I am puzzled about the result in Fig. 3c. Why would the error in sO₂ change with the concentration of the probe if the signal is linear with fluence and it is measured at the same point (no differences in spectral coloring)?

As described previously, this is an error in spectral unmixing, not due to nonlinearity of the monomeric ICG PA signal, which Supplementary Fig. 6c shows to be quite linear at the concentrations used. This point has since been clarified in the

manuscript. In the case of increasing error from increasing probe concentration, there is an increase in the monomeric ICG peak, which is centered at ~800 nm and extends down to ~750 nm (see Supplementary Fig. 1). Because this is near the local maximum for HHb at 760nm, some of this signal is incorrectly assigned to hemoglobin rather than ICG during unmixing, resulting in a much greater SO₂ underestimation error at 0.9 OD than what is obtained at a reduced concentration of 0.3 OD, where the aforementioned confounding effect becomes less significant. This effect is also evident in the only 1.4-fold increase observed in ICG signal when the probe concentration is tripled (i.e., 0.3 to 0.9 OD at 50% SO₂; see Fig. 3d). In this case, less PA signal is assigned to ICG during unmixing, so the increase in ICG signal is lower in magnitude than it should be given the concentration increase.

11. The authors mention that cytotoxicity of PAttrace at high concentrations may be associated to the lipid coating. This should be assessed e.g. by considering only liposomes or PAttrace probe with no targeting moiety.

We thank the reviewer for this suggestion. We used HUVEC endothelial cells as a model of normal human cells to evaluate cytotoxicity of PAttrace, empty liposomes, and ICG dye alone. This approach was also suggested by Reviewer 2. Normal cells were chosen for two main reasons: (1) they are generally less robust as compared to cancer cells, and (2) it is important to evaluate potential “bystander” cytotoxicity to normal cells. These data are shown in Supplementary Fig. 12 for PAttrace and Supplementary Fig. 13 for empty liposomes and ICG dye alone, while a summary of these results is also now included in the revised manuscript. Briefly, PAttrace exhibited a cytotoxic effect in HUVEC cells only at the highest concentration. No cytotoxicity was observed for empty liposomes at concentrations up to 2.7x the highest PAttrace concentration. With ICG, HUVEC cells showed cytotoxicity at high ICG concentrations, as shown in Supplementary Fig. 13. These data indicate that the observed minor cytotoxic effect at high PAttrace concentrations might be associated with a very high loading of ICG dye molecules inside PAttrace.

12. I would recommend to make experiments that enable characterizing the extravasation of PAttrace and how this compares with that of targeted PAttrace. For example, in Ermolayev et al. European Radiology 26 (6), 1843-1851 (2016), the authors use a real-time imaging system to visualize the extravasation of liposomal ICG. I think this is very interesting and doable with the real-time imaging system available to the authors.

Thank you for bringing this very relevant prior study to our attention. We have since added it as a reference for PA imaging of non-J-aggregated ICG encapsulated in liposomes. As you point out, investigators in this study noted that their PA images “exhibited initial signs of extravasation into the tumour mass.” And although we would certainly love to explore such initial signs of extravasation in our tumor model, there are two significant differences between the Ermolayev *et al.* study and ours that make such an investigation very challenging. Firstly, they used a significantly different PA imaging system that is custom-built but similar to the commercially available Endra Nexus platform. Such systems provide very clear visualization of vasculature at relatively shallow ROIs; they are not, however, intended for the type of whole-body imaging for which our iThera inVision system is best suited. Secondly, to be within the aforementioned system’s imaging “sweet spot,” the investigators used a more shallow-lying subcutaneous tumor model, not a deeper-lying orthotopic model, such as was used in our study. Nonetheless, we feel your suggestion is an excellent one, and thus we plan to explore extravasation in our tumor model in future work, a point that has now been stated in the Discussion.

13. I am also missing a characterization of the photo-stability of the PAttrace with and without targeting moiety and how this compares with monomeric ICG.

This is an excellent point, and we apologize for not addressing it in our original submission. We have since assessed the thermal stability of targeted PAttrace (i.e., FR α -PAttrace) to over 1,000 laser pulses near the ANSI fluence limit for skin (i.e., 20 mJ/cm²) to create a “worst-case” heating scenario. As is now shown in Supplementary Fig. 11, targeted PAttrace presents with the same final spectrum – both in shape and in magnitude – as it presents initially and before significant laser-pulse irradiation. We also conducted the same photostability for monomeric ICG (see Supplementary Fig. 11), which showed signs of photobleaching, as evidenced by a final PA spectrum that was of lesser magnitude than its initial spectrum. Additionally, these spectra look very similar to the UV-Vis-NIR spectrum for PAttrace (see Fig. 2c). Supplementary Fig. 4a also shows stability of PAttrace exposed to fluences in excess of 20 mJ/cm².

REVIEWERS' COMMENTS

Reviewer #1 (Remarks to the Author):

The authors have fully addressed my concerns in the first-round review, with new experimental data and detailed discussions. I think the revised version is suitable for publishing in Nature Communications.

Reviewer #2 (Remarks to the Author):

The paper can be accepted for publication

Reviewer #4 (Remarks to the Author):

The revised manuscript on photoacoustic imaging using a novel nanoparticle: liposome-encapsulated ICG J-aggregates. The authors have done a nice job addressing the comments and suggestion from the first round reviewers. In particular, the nanoparticle fabrication process, the cell toxicity, and the PA signal analysis are described in details, which substantially improved the manuscript rigor. I believe this work will generate strong interest from the general photoacoustic imaging community as well as the optical imaging contrast field. I suggest acceptance of this new work aiming to push the clinical translation of photoacoustic molecular imaging. I have two minor comments for the authors to consider when finalizing the manuscript.

1. I suggest the authors to clarify in the manuscript what is the most important contribution of this new work to photoacoustic molecular imaging. Spectral unfixing is an outstanding issue in photoacoustic imaging, especially in deep tissue, as clearly and correctly pointed out by the original reviewer 3. Although the new J-aggregates seems to be better than monometric ICG in terms of its spectral stability and absorption peak, this new contrast agent does not fundamentally address the inaccurate nature behind the spectral unmixing, which is the spectral coloring. Unless a completely different imaging mechanism such as the recent photoswitching based contrast, it is likely that the spectral unmixing method used in this manuscript will be sensitive to a number of confounding factors, which will be even worse when, as intended, it is applied on humans with deeper penetration and lower signal to noise ratio.

2. I suggest the authors to acknowledge/discuss the further improvement on photoacoustic imaging in order to maximize the impact of the new contrast. The photoacoustic imaging quality presented in this work is somewhat subpar to the state-of-the-art photoacoustic whole-body small animal imaging reported by other groups, and thus the presented molecular imaging performance (such as the unmixing accuracy) is likely underestimated. It is understandable that the authors are relying on the commercial PA system which does not necessarily provide the most advanced imaging quality or data analysis capability. As the authors intend to translate the new contrast for clinical applications, the imaging quality needs to be improved.

Junjie Yao

Clinically translatable quantitative molecular photoacoustic imaging with liposome-encapsulated ICG J-aggregates

Cayla A. Wood, Sangheon Han, Chang Soo Kim, Yunfei Wen, Diego R.T. Sampaio, Justin T. Harris, Kimberly A. Homan, Jody L. Swain, Stanislav Y. Emelianov, Anil K. Sood, Jason R. Cook, Konstantin V. Sokolov, Richard R. Bouchard

We would like to thank the editor for arranging the review of our manuscript and the referees for their helpful comments. We have addressed all reviewer comments and suggestions and feel that the quality and clarity of our manuscript has improved as a result. Reviewer comments are provided verbatim in bold text, and our reply to each comment is given immediately following.

Reviewer #1:

The authors have fully addressed my concerns in the first-round review, with new experimental data and detailed discussions. I think the revised version is suitable for publishing in Nature Communications.

We thank the reviewer for their input and support to publish our manuscript.

Reviewer #2:

The paper can be accepted for publication

We thank the reviewer for their input and support to publish our manuscript.

Reviewer #4:

The revised manuscript on photoacoustic imaging using a novel nanoparticle: liposome-encapsulated ICG J-aggregates. The authors have done a nice job addressing the comments and suggestion from the first round reviewers. In particular, the nanoparticle fabrication process, the cell toxicity, and the PA signal analysis are described in details, which substantially improved the manuscript rigor. I believe this work will generate strong interest from the general photoacoustic imaging community as well as the optical imaging contrast field. I suggest acceptance of this new work aiming to push the clinical translation of photoacoustic molecular imaging. I have two minor comments for the authors to consider when finalizing the manuscript.

We thank the reviewer for their input and support to publish our manuscript.

1. I suggest the authors to clarify in the manuscript what is the most important contribution of this new work to photoacoustic molecular imaging. Spectral unfixing is an outstanding issue in photoacoustic imaging, especially in deep tissue, as clearly and correctly pointed out by the original reviewer 3. Although the new J-aggregates seems to be better than monometric ICG in terms of its spectral stability and absorption peak, this new contrast agent does not fundamentally address the inaccurate nature behind the spectral unmixing, which is the spectral coloring. Unless a completely different imaging mechanism such as the recent photoswitching based contrast, it is likely that the spectral unmixing method used in this manuscript will be sensitive to a number of confounding factors, which will be even worse when, as intended, it is applied on humans with deeper penetration and lower signal to noise ratio.

We agree with the reviewer that photoswitching-based contrast may be able to provide improved PA imaging contrast in future applications. Thus, we have now explicitly acknowledged this exciting new platform in the

Introduction of the manuscript: *“Additionally, bacterial phytochromes have been used to provide improved PA contrast due to their ‘photoswitchable’ optical absorption³¹.”* In the Discussion, we have also mentioned the potential need to address spectral coloring, particularly for deeper-lying (clinical) ROIs: *“However, it must be noted that unmixing accuracy is still negatively affected by spectral coloring, and thus application of local-fluence correction (i.e., not simply surface-fluence correction, as was applied in this work) may be necessary for the relatively deep ROIs likely to be encountered for in-human applications.”*

2. I suggest the authors to acknowledge/discuss the further improvement on photoacoustic imaging in order to maximize the impact of the new contrast. The photoacoustic imaging quality presented in this work is somewhat subpar to the state-of-the-art photoacoustic whole-body small animal imaging reported by other groups, and thus the presented molecular imaging performance (such as the unmixing accuracy) is likely underestimated. It is understandable that the authors are relying on the commercial PA system which does not necessarily provide the most advanced imaging quality or data analysis capability. As the authors intend to translate the new contrast for clinical applications, the imaging quality needs to be improved.

We agree with the reviewer that improved imaging hardware may be required to achieve optimal results, particularly in a clinical setting. Thus, we have added verbiage to the Discussion to reflect this: *“Additionally, further improvements in the imaging sensitivity of clinical PA transducers, which typically have a reduced receive-angle (e.g., linear array) compared to the MSOT inVision’s arc array, may be required to achieve clinical performance commensurate with the preclinical results shown herein.”*